

# An examination of land use impacts of sea level rise induced flooding

Jie Song[1], Xinyu Fu[1], Yue Gu[1], Yujun Deng[1], Zhong-Ren Peng[1]

[1]Department of Urban and Regional Planning, College of Design, Construction, and Planning, University of Florida, POB 115706, 32611, USA

*Correspondence to*: Zhong-Ren Peng (zpeng@ufl.edu)

**Abstract.** Coastal regions are under intense development because of their biodiversity and economic attractiveness. Meanwhile, these places are highly vulnerable to coastal hazards that are associated with sea level rise. Continuing urban development in these coastal areas that are prone to be flooded increasingly poses unnecessarily risk to their residents. While overwhelming efforts have been made to investigate coastal land use changes, few studies have simultaneously explored

urban growth dynamics and its interaction with the coastal hazards. This paper applied the cellular automaton-based SLEUTH model to calibrate historical urban growth pattern from 1974 to 2013 in Bay County coastal areas, Florida. Three scenarios of urban growth---historical trend, compact development, and urban sprawl---up to 2080 were predicted by applying the calibrated SLEUTH model. To assess the effects of different policies, we developed three excluded layers---no regulations, flooding-risk mitigation, and conservational/agricultural land protection---and evaluated how different urban

growth scenarios were oriented under these policies. Eventually, flooding maps were overlaid with future urban areas, and the exposure of different urban growth patterns to sea level rise induced flooding was examined. The findings suggest that if coastal cities expand in a compact manner, areas vulnerable to flooding will increase compared with historical trend and urban sprawl scenarios. With respect to policies, if no regulations are implemented, on average the flooded area in 2080 would be more than 25 times under flooding-risk mitigation. The joint model can serve as a decision support tool to assist

city officials, urban planners, and hazard mitigation planners in making informed decisions. The visualization results can be also useful in public outreach regarding coastal communities' increasing risk to flooding enhanced by sea level rise.

## 1 Introduction

Coastal areas are the most intensively exploited places where urban expansion largely alters natural landscape. As land-sea interfaces, however, coastal regions are featured by various conflicts between anthropogenic pressures and natural forces.

Moreover, such conflicts have been exacerbated in recent years. While coastal zones increasingly attract population and investments, their communities become more aware of intensified frequency of natural incidents. Climate change is likely to contribute to intensified hurricanes and floods (Hsu, 2014), rising sea level (IPCC, 2013), and other coastal hazards. More specifically, coastal flooding, land submergence, and saltwater intrusion may be worsened by sea level rise (Nicholls & Cazenave, 2010). According to the fifth assessment report published by the Intergovernmental Panel on Climate Change

(IPCC) in 2013, by 2100 approximate 70% coastlines will experience rising sea levels. Yet, different stakeholders---



residents, tourists, developers---still compete for limited coastal resources, and the competition has been even more intense in recent years. Coastal areas attract more residential projects and infrastructure development; oil and natural gas are extensively extracted in offshore regions (Felsenstein & Lichter, 2014). Since coastal zones are both vulnerable low-lying places and the battleground of conflicting interests, coordinating land use, hazard mitigation, and different interests has

become everlastingly important. The coordination, therefore, calls for effective tools to assist in the formulation of coastal management plans. Moreover, a pivotal part of management plans is spatial planning, which can be oriented by identifying coastal land use/land cover changes.

Various techniques have been applied to detect patterns in land use/land cover changes. The class of Cellular Automaton (CA) models receive more attention around the world due to their simplicity and effectiveness in capturing complex urban

development mechanism (Akın, Clarke, & Berberoglu, 2014). The operationalization of CA in modelling urban phenomena was first introduced by (Clarke, Hoppen, & Gaydos, 1997). In their model each cell had a state which updated at every time step according to a predefined set of transitional rules. These rules took into account the current state of a cell and its neighbours as well as environmental constraints. A wide range of models and software packages have been developed since the introduction of CA. Santé et al (2010)   evaluated thirty-three CA related models and concluded that two widely applied

models were SLEUTH and another package developed by White and Engelen (2000) . SLEUTH has been adopted by urban researchers all over the world since 2000 (Project Gigalopolis, 2016). Its popularity among urban growth modelers is largely due to the following aspects: it is free and supported by a technical forum; it only requires six data inputs (i.e., slope, land use, exclusion, urban extent, transportation, and hill shade); it follows a well-documented and routine historical calibration process (Sekovski et al., 2015); and it is very computationally efficient due to recent updates. Recently, SLEUTH has been

improved by the incorporation of external information during calibration. Rienow and Goetzke (2015) used support vector machine to enhance the predicting power of SLEUTH by developing probability-based excluded layers for future scenarios. Sakieh et al. (2015) applied multi-criterion evaluation method to develop suitability surface as excluded layers.

Based on historical urban expansion patterns, future land use could be simulated via scenario-based SLEUTH and other CA models. By applying a constrained CA model, Hansen (2010) simulated future land conditions under different emission

scenarios developed by the IPCC (2013) and concluded that considerable areas in Aalborg would be exposed to potential flooding. Although the author incorporated adaptation strategies in his model, Hansen (2010) suggested that more radical strategies, such as relocation and managed retreat, may be evaluated in future simulations. Likewise, Sekovski et al. (2015) assessed the impacts of coastal flooding upon various patterns of urban growth using the SLEUTH model. Inouye et al. (2015) applied a comparative modelling approach to validate the importance of zoning in urban growth models. By applying

the Dynamic EGO software, they stated that development in ecological-economic zones may heighten the exposure of future urban settlements to land sliding, intensified precipitation, sea level rise, and other coastal hazards. Their results provide actionable information to decision makers to develop relocation, planned retreat, and other adaptation policies. In addition, some other researchers highlighted the importance of incorporating zoning information in SLEUTH applications. Akin et al. (2014) used current zoning maps as excluded layers and evaluated hindcasting-based calibration in SLEUTH. Onsted and



Chowdhury (2014) employed a historical zoning map in their SLEUTH model and suggested that land use zoning strongly influenced urban growth in their study region. While the majority of SLEUTH applications are centered on assessing prospective urban sprawl, compact development, and other developmental patterns, the investigation of future land use exposure to coastal flooding received less attention. To the best of our knowledge, only two studies (Garcia & Loáiciga, 2014; Sekovski et al., 2015) attempted to couple land use prediction with marine flooding maps using SLEUTH. The combination of SLEUTH simulation and flooding hazard maps, however, should be prioritized in coastal land use modelling to better inform coastal spatial management (Jeffrey A. Onsted & Chowdhury, 2014). Therefore, this study aims to evaluate the extent to which different urban growth patterns may be exposed to sea level rise induced flooding. Specifically, two primary research questions of this study are:

1. How would different urban growth patterns increase regional vulnerability to sea level rise induced flooding?
2. Would future land use zoning and flooding mitigation plans help steer prospective development away from flood-prone regions?

This paper is organized as follows. Section 2 describes the study area and data collection. Following this, section 3 illustrates the modelling framework and outlines major steps for calibration, prediction, as well as floodplain generation. In section 4 we present the calibration coefficients and discuss forecasting maps that were overlaid with flooding maps. Finally, section 5 offers a brief conclusion and provides outlook for future research.

## 2 Study area and data description

### 2.1 Study area

The study region is located in the north-western Florida and has long shorelines which form four bays along the Gulf Coast (Figure 1). It covers Bay County and some areas of Washington and Walton County. Its topographical features and past land use development render this region extremely susceptible to storm surges and hurricanes. For instance, considerable residential and commercial buildings encroached upon seafront areas in Panama City due to the absence of land use regulations in the past; as a result, urban growth largely occurs in low lying and flood-risk zones (Bay County Online, 2016). Historically, the study area has been hit by seventeen hurricanes since 1877 (Hurricanecity, 2015). Among these incidences, Hurricane Eloise in 1975 led to $23.1 million damages in structures, seawalls, and patios (Shows, 1978).

FIGURE 1 ABOUT HERE

The economic activities and population profile of the study area increase its vulnerability to coastal hazards as well. Bay County highly relies on tourism related industries; specifically, restaurant and real estate are major industry sectors in Panama City. In Bay County the total spending on tourism as of June 2015, which was $121 million, had doubled since 2008 (Bureau of Economic and Business Research, 2015). Bay County has 10,222 firms, of which approximate 90% are small



businesses which are highly susceptible to environmental disasters (Howe, 2011; Runyan, 2006). The overall population of Bay County is 168,852, around 30% of which come from two major coastal cities: Panama City and Panama City Beach (U.S. Census Bureau, 2015).

## 2.2 Urban change and zoning

5 Figure 2 displays historical urban growth for the study area. It indicates that urban extent expanded primarily in the southern part of Bay County. As shown in Figure 2, urban development largely conforms to historical trends. Substantial commercial and residential developments occur in two major cities: Panama City Beach and Panama City. In addition, a large piece of land in the north region is zoned for residential uses. This information suggests that local governmental and planning agencies have taken measures to encourage inland developments. There exists, however, an apparent discrepancy between 10 zoning and urban growth. It is noted that substantial built up areas have appeared in the Town of Fountain and Youngstown. Yet, such a pattern is not seen in current zoning where only some areas are designated for residential use in these two towns. This inconsistence suggests that a comprehensive understanding of past land use changes is require to better inform prospective land use zoning.

15 FIGURE 2 ABOUT HERE

## 2.3 Data description

Land use and sea level rise data were prepared for this study. The land use data were comprised of five remotely sensed images on 1974, 1995, 2004, 2007, and 2013; these data were obtained from the Florida Geographic Data Library. These data sets were categorized into nine level-one land cover classes, among which built-up land was coded as one, according to 20 the Florida land use, cover and forms classification system published by the Florida Department of Transportation (1999). Flooding hazards and zoning information were also collected. Two Flood Insurance Rate Maps (FIRM), developed by U.S. Federal Emergency Management Agency, were downloaded from the Florida Geographic Data Library. Current zoning map and the comprehensive plan for the study area were obtained from the online GIS websites of Bay County and Washington County. Land use zoning specified exactly the degree to which different land uses were allowed for urban development. 25 According to the comprehensive plan, the zoning regulations are shown as follows:

-Residential uses were prohibited in the zone of conservation/preservation, and impervious areas must be no more than 5% in this zone;

-In the zone of conservation/habitation impervious surface must be no more than 50%;

-In the zone of conservation/recreation impervious coverage must be no more than 10%; and

30 -In the zones of agriculture/timberland and agriculture impervious areas must be no more than 10% and 25% respectively;





Finally, the data of projected sea level rise and sea surface temperature were collected from the fourth assessment report published by the IPCC (2007). Table 1 displays detailed information regarding all data sets.

TABLE 1 ABOUT HERE

**3 Method**

**3.1 An induction to SLEUTH**

**3.1.1 Background**

SLEUTH is a packed C language-based source code that was developed by Dr. Keith C. Clarke at the Department of Geography, University of California, Santa Barbara. The source code is freely available through its official website entitled
"Project Gigalopolis" (http://www.ncgia.ucsb.edu/projects/gig/index.html).

SLEUTH is comprised of two modules: Urban Growth Model and Land Cover Deltatron Model. The Urban Growth Model mainly focuses on the urban/non-urban dynamics. It is more frequently applied among SLEUTH users than the Land Cover Deltatron Model which investigates changes among different land cover classes. The Urban Growth Model is, therefore, a primary focus of this work. As mentioned, SLEUTH is a CA-based program which only relies on five inputs: urban,
transportation, slope, hillshade, and exclusion, and thus it is moderately data driven. The family of CA models have gained more popularity than other modelling techniques among urban modelers. CA models are advantageous over other counterparts due to their spatial explicitness, flexible transitional rules, powerful performance with large data sets (Wagner, 1997), and easy integration with Geographical Information System (Santé et al., 2010). Furthermore, among numerous CA models SLEUTH was selected for this research because of the following considerations. First, SLEUTH employs excluded
layers as probability maps which specify developmental potentials over a study region (where the cell value of zero represents an attracting point for development, and 100 or higher reflects that development is strictly prohibited). Such a functionality makes SLEUTH an excellent platform for scenario-based studies (Leão, Bishop, & Evans, 2004; Jeffrey A. Onsted & Chowdhury, 2014). Second, SLEUTH uses five parameters---dispersion, breed, spread, road gravity, and slope--- to establish transition rules which determine whether or not a cell is urbanized. Finally, the calibration process applies a
"brute force" approach and is scientifically sound for regional studies (Jeffrey A Onsted & Clarke, 2011).

**3.1.2 SLEUTH workflow**

SLEUTH is a scale-independent CA model that updates the binary state of each cell per growth cycle. A growth cycle is normally one year and consists of four steps: spontaneous growth, new spreading centres, edge (organic) growth, road-influenced growth (Clarke et al., 1997). These transitional rules are controlled by one or more parameters that were
mentioned previously. Each parameter, which has a value range of 1 – 100, is dimensionless and can be compared in terms



of their contributions to overall growth. Specifically, the dispersion factor determines the probability by which a cell will be randomly selected for urbanization. The breed factor determines the likelihood by which a newly formed urban cluster will start its own growth cycle. The spread factor controls how likely outward growth will happen surround an existing settlement. The road gravity factor demonstrates the influence of road systems upon urban growth by attracting new

settlements which are within a certain distance of a road. Finally, the slope factor determines how likely a cell with steeper slope will be urbanized. Table 2 summarizes the relationships between transitional rules and five parameters.

TABLE 2 ABOUT HERE

Main workflow of a SLEUTH application includes input compilations, calibration based on historical urban growth, and predictions. In Urban Growth Model, at least four maps of different dates which show discernable urban changes are required. Two road networks of different periods and one slope map with percentage rise are additional date sets. Finally, a hillshade map is used to enhance visualization performance, and water bodies can be embodied in this map. The goal of calibration is to select a combination of parameters which best simulate historical urban changes. This process, however, is

enormously time-consuming if all combinations (up to 10 billion) are evaluated. Therefore, a four-stage calibration---coarse, fine, final, and derive----is applied to reduce computational time yet retains acceptable accuracy. Eventually, predictions with 100 Monte Carlo runs are conducted using the best-fit parameters. Full descriptions regarding model inputs, calibration, and prediction can be found in Project Gigalopolis (http://www.ncgia.ucsb.edu/projects/gig/Imp/implement.htm).

        The workflow for this research was organized in four phases: 1) the preparation of inputs layers, 2) the calibration

in SLEUTH environment, 3) the prediction of urban growth up to 2080 under the combined scenarios of different developmental patterns and excluded layers, and 4) the overlaying of urban growth estimates and 500-year flooding maps that were induced by sea level rise. The overall research framework is displayed in Figure 3. Overall, the workflow was divided into two major tasks: SLEUTH urban growth model and sea level rise induced flooding maps.

25                              FIGURE 3 ABOUT HERE

**3.2 Urban growth model**

**3.2.1 Land use/cover related layers**

All input data were processed in ArcGIS 10.3. Five land use maps in vector format were converted into grid raster files using the nearest neighbourhood method for spatial resampling. Figure 4 shows urban changes of the study area from 1974 to

30  2013.

FIGURE 4 ABUOT HERE



Due to the absence of reliable records of transportation networks, only two most recent road maps were used. Because local roads may have very limited influence upon urban growth, only main arteries were extracted from original line files according to the MAF/TIGER Feature Class Code. These polylines were then converted into raster files using the nearest

neighbour resampling method. Slope and hillshade were finally generated from National Elevation Dataset using spatial analyst tool in ArcGIS.

### 3.2.2 The creation of E1 excluded layer

An excluded layer reflects the urbanization probabilities of cells. Its cell values range from 0 (unaffected) to 100 (entirely excluded) (Akın et al., 2014). Many publications have investigated how to use excluded layers to enhance calibration

accuracy (Rienow & Goetzke, 2015; Sakieh et al., 2015) and evaluate policy scenarios (A. S. Mahiny & Clarke, 2012). Onsted and Clarke (2012)  and Akin et al. (2014) recommended that different excluded layers be used in calibration and prediction. The excluded layer for calibration is suggested to be at minimal restrictions in order to obtain more precise results of urban growth, according to Akin et al. (2014). Hence, this excluded layer (E0) only covers water bodies where urban developments are unrealistic (Figure 4). For prediction phase, however, two excluded layers were applied to represent

flooding-risk mitigation (E1) and conservational/agricultural land protection (E2).

E1 attempted to assess how likely urban growth appeared in Special Flood Hazard Area (SFHA) which may be flooded by 100-year flooding. Mandatory flood insurance must be purchased by land owners in these areas. Raising construction costs in high-risk regions may partly inhibit vulnerable urban growth. Therefore, E1 could represent a scenario which guides urban development towards less flood prone areas. In order to avoid arbitrarily assigning values to excluded layers, Onsted et al.

(2014) developed an approach to use historical zoning maps to calculate these values, which was adopted in the creation of E1 excluded layer. Essentially, their method relies on the growth rates whereby urban development occurs in each zone in order to estimate weighted values. To retrieve past growth information, we selected 1996 Flood Insurance Rate Map (FIRM) as a reference layer (Figure 5) and calculated the area of SFHA and non-SFHA zones as well as the amount of new urban areas from 1995 to 2013 in these zones.

FIGURE 5 ABOUT HERE

Next, annual rate of urban growth in each zone was determined by Eq. (1):

$$g_n = 1 - ((1 - (\frac{G_n}{Z_n}))^{(1/T)})$$  (1)

where $G_n$ is the total actual urban growth in zone $n$ (1 is SFHA and 2 is non-SFHA zone) from 1995 to 2013,

$Z_n$ is total area of zone $n$ according to the 1996 FIRM, and



$T$ is the number of years, i.e., 18 years.

Finally, the growth rates were used to generate excluded value in the SFHA zone by Eq. (2):

$$E_{SFHA} = 100(1 - (\frac{g_1}{g_2}))$$
(2)

where $g_1$ and $g_2$ denote growth rates in SFHA and non-SFHA zone respectively. Table 3 indicates that growth rate in low-

risk areas was approximate three time in SFHA zone, suggesting that mandatory flooding insurance constrained urban

expansion in vulnerable regions.

TABLE 3 ABOUT HERE

Finally, E1 layer was created based on the 2015 FIRM and represented a managed growth plan that accounted for moderate

protection from flooding risk (Figure 6).

FIGURE 6 ABOUT HERE

### 3.2.3 The creation of E2 excluded layer

The 2020 Bay County Comprehensive Plan was published in 2009 and represented the most recent managed growth option

for the study area. Hence, excluded values in the E2 layer were weighted according to this plan and modified based on the

work of (Akın et al., 2014). Specifically, a cell value of 100 was assigned to water bodies, 95 to the

conservation/preservation zones, 50 to the conservation/recreation and agriculture/timberland areas, 25 to other agricultural

areas and conservation/habitation zones, and 0 to all other areas (Figure 7).

FIGURE 7 ABOUT HERE

Eventually, all these raster files were resampled at a spatial resolution of 30 m * 30 m which is adequately high since the

resolutions of most SLEUTH applications are in the range of 10-100m (Akın et al., 2014). These raster files with 1972 rows

* 2383 columns were then exported as grayscale GIF images and imported into SLEUTH program.

### 3.2.4 Model calibration

As mentioned, urban-growth parameters were calibrated based on the "brute force" technique where four calibration stages---
-coarse, fine, final, and derive—were followed. Increasingly higher image resolutions were typically used from coarse to

derive calibrations for computational efficiency (Akın et al., 2014; Chakraborty, Wilson, & Kashem, 2015; Rafiee, Mahiny,

Khorasani, Darvishsefat, & Danekar, 2009). Different resolutions, however, may be problematic and lead to a biased



estimation of growth patterns (Dietzel & Clarke, 2007; Sekovski et al., 2015). Thus, a consistent resolution of 30 * 30 m was employed during the whole calibration.

In each calibration phase, several Monte Carlo iterations were run to account for the uncertainty associated with parameter estimations. A general strategy of identifying "best-fit" parameters is to shrink the range of parameters during each stage.

Hence, in the coarse calibration four Monte Carlo iterations were conducted, and the widest range of parameters, 0 to 100, were evaluated with an increment of 25 at a time. The goodness-of-fit of models were assessed by thirteen metrics, the majority of which were least-square regression scores between simulated urban components (e.g, increased urban pixels and clusters) and actual counterparts. SLEUTH scholars, however, largely debated the optimal metric which could best describe model performance. Most disputes centered on using single metric or a combination of several indicators. Dietzel and Clarke

(2007)  developed a composite metric, known as the Optimal SLEUTH Metric (OSM). OSM is the product of the compare, population, edges, clusters, slope, X-mean, and Y-mean metrics and was applied in this work to narrow parameters ranges after each stage.

Seven Monte Carlo iterations with narrower parameter ranges were employed in the fine stage. Further refined parameter ranges with nine Monte Carlo iterations were next tested during the final calibration. This whole process took around one-

month CPU time and was conducted using the standard 3.0 SLEUTH model executed in the Cygwin UNIX windows compiler. This three-stage calibration generated five candidate parameters; however, this set may be biased due to the self-modification nature of SLEUTH. Therefore, a "derive" calibration with the candidate set were performed with 100 Monte Carlo iterations.

### 3.2.5 Model prediction

Three approaches have been widely applied in model prediction. The first is to adjust growth parameters which affect the way urban growth evolves (i.e., infilling or outward dispersion) (Leao, Bishop, & Evans, 2004; Rafiee et al., 2009; Sekovski et al., 2015). The second is to adjust growth-resistance levels in excluded layers (Jantz, Goetz, Donato, & Claggett, 2010; Sekovski et al., 2015) or apply distinct excluded layers as different policies (Akın et al., 2014). The last one, less frequently used than the first two methods, is to alter self-modification parameters which control overall growth rates (Yang & Lo,

2003). This research used a combination of the first two approaches, and the results were overlaid with sea level rise induced flooding maps. The urban growth was predicted up to 2080, and the exposure to flooding under different growth patterns and policies were analysed in 2030 and 2080.

As in widespread use in similar studies, this work first simulated three growth patterns: Historical Growth (HUG), Urban Sprawl (USG), and Compact Development (CUG). HUG assumed that prospective urban extent expanded at existing growth

rates. Five parameters remained unchanged over the forecasting period. USG resulted in more scattered urban communities which appeared in suburbs and along transportation corridors. While urban sprawl caused various socioeconomic issues and was discouraged in contemporary planning, it may benefit vulnerable coastal communities by steering developments away from flood prone areas. In the model sprawling growth was controlled by the dispersion, breed, and road gravity, so



increasing these parameters produced more dispersed growth. On the contrary, CUG greatly relied on the expansion of current settlements. Compact development is apparent in the study area and many other populated coastal regions. In SLEUTH compact development was positively associated with spread. In addition, decreasing road gravity inhibited new growth along corridors and thus contributed to compact urban form.

These growth patterns were next simulated under two policy scenarios: flooding-risk mitigation (E1) and conservational/agricultural land protection (E2). E1 restrained growth in low-lying areas and served as an adaptation strategy to sea level rise. Alternatively, E2 reflected how future city expansion may be impacted by land use zoning which represented a strong predictor of urban growth in Florida (Jeffrey A. Onsted & Chowdhury, 2014). Finally, these urban growth predictions were coupled with sea level rise induced flooding, whose methodology was briefly introduced in the next
section.

### 3.3 Flooding maps

The detailed methodology for generating sea level rise induced flooding were developed by (Hsu, 2014). In the hurricane model, the effects of rise in sea level and Sea Surface Temperature (SST) were considered in two stages. First, increase in SST heightened hurricane central pressure (Knutson & Tuleya, 2004). Increased central pressure and other parameters were
next used to calculate projected surge height using the Surge Response Function (SRF) developed by (Irish, Resio, & Cialone, 2009). Based on this information, a hypothetical hurricane was projected to make landfall at a place where it caused the most damages to coastal areas and resulted in a 500-year flood. Second, the projected surge height for the study region was adjusted by local sea level rise (Udoh, 2012). Two extreme sea level rise scenarios were considered, as shown in Table 4. A1F1 corresponded to the highest level of global greenhouse gas emissions (IPCC, 2007). Global data were finally
adjusted according to local marine conditions in Panama City.

TALBE 4 ABOUT HERE

The surge heights were calculated in numerous SRF stations which were defined along the coastline of study area. SRF
zones associated with each station were delineated, and each zone had a height value. Eventually, flooding areas were identified by comparing surge heights and local elevation data.

### 4 Results and discussion

### 4.1 Model calibration

The multi-stage calibration process generated the following parameters: 71 (dispersion), 92 (breed), 70 (spread), 3 (slope),
and 35 (road gravity). High values of the first three parameters suggest that past several decades have seen apparent sprawl and growth surrounding established settlements. On the contrary, low slope value is intuitively reasonable since the case





study region barely has mountainous areas, and therefore slope is not a limiting factor. The impact of major roads is rather limited partly because the road network has remained relatively stable since 1980s.

**4.2 Model prediction**

Similar studies have suggested that a sensitivity analysis be conducted before prediction in order to identify the most
significant parameter (Sekovski et al., 2015). The assessment was conducted by subsequently setting each parameter as 80 and keeping the others as the lowest value of 1 and running predictions up to 2030. The results indicate that the spread coefficient has the greatest impact upon future urban expansion, resulting in a 13.99% increase in urban areas up to 2030.
Urban sprawl and compact development can be characterized by different sets of parameters. Specifically, urban sprawl is referred to as scatteredly formed settlements and developments along major transportation networks. Conversely, compact
developments appear surrounding existing urban areas. Therefore, two scenarios, urban sprawl (USG) and compact development (CUG), were developed according to the following criteria.

1.   The dispersion, breed, and road gravity coefficients were increased and decreased by 25 in USG and CUG respectively.
2.   The spread coefficient was risen and lowered by 10 in USG and CUG respectively. Since this parameter was much
15        more influential than the others in terms of affecting urban growth, 10 was selected as an adjusting value in two scenarios.
3.   As its impact was quite marginal, slope parameter remained unchanged across all scenarios. Table 5 summarizes different sets of parameters for three scenarios.

20                                      TABLE 5 ABOUT HERE

By applying different parameter combinations, the SLEUTH model generated various maps which showed the probability of each cell being urbanized. These maps could be then converted to urban / nonurban results by a cut-off probability value. A justified approach to identify the reasonable cut-off value is to assess the histogram frequency of probability (Dezhkam,
Amiri, Darvishsefat, & Sakieh, 2013; Rafiee et al., 2009; Wu et al., 2008). After evaluating forecasting maps in 2080, we found that there was a steep increase of urbanized cells around the probability value of 90. The cut-off value of 85, accordingly, was selected to determine whether a cell was converted into urban.
The results show that, under no land use regulations, city areas increase substantially under all scenarios. Urban region, for instance, expands up to 826 km$^2$ in 2080 under historical growth. Similar patterns can be seen in alternative growth scenarios
as well, as shown in Figure 8 a. Under stricter restrictions, the simulations with E1 excluded layer (flooding-risk mitigation) generate the smallest increase in urban extent from 2013 to 2080 (Figure 8 b). In addition, the growth curves which gradually level off from 2013 indicate a constantly decreasing growth rate. Under compact growth, which shows the most rise in urban areas among three scenarios, the city region expands by 15% within seven decades (Figure 8 b). This is intuitively





reasonable because the flooding map exerts a heavy constraint on undeveloped lands, and therefore new developments largely appear surrounding existing settlements. However, urban growth under conservational/agricultural land protection (E2) has a similar pattern as simulations with no regulations (Figure 8 c). The growth rate in compact development, for instance, reaches the peak (19%) at 2044 and then gradually levels off. Yet, land use zoning does have an impact upon the

amount of new urban areas. The simulated urban area in 2080 with historical growth pattern is 709 km$^2$ (Figure 8 c), only 85% of that under no restrictions (Figure 8 a). Overall, the results are consistent with similar findings (Sekovski et al., 2015): spreading development from existing coastline areas is the leading force behind land use changes, and the changes are also driven by dispersion, breed, and road network but less associated with slope factor.

10                                        FIGURE 8 ABOUT HERE

Figures 9 - 11 show predicted urban growth up to 2080 under different excluded layers. These illustrations further depict urban expansion trends indicated by the growth curves shown in Figure 8. In other words, historical growth and urban sprawl share similar development patterns where the majority of increased urban cells appear under less strict land use regulations.

Moreover, a considerable portion of urban development would be steered towards flooding zones if no land use policy is implemented, as indicated by the highlighted boxes in Figures 9 – 11.

                                          FIGURE 9 – 11 ABOUT HERE

**4.3 The exposure of urban growth to flooding risk**

Figure 12 shows 500-year flooding maps that would be exacerbated by sea level rise in 2030 and 2080.  A vast region along the West, North, and East Bay would be flooded, and the areas immediate to the West and North Bay would be even more susceptible in 2080. As shown in Table 6, the total inundated area in 2080 would be more than 10 times in 2030. Additionally, affected areas with the flooding depth over 3 m would increase exponentially from 2030 to 2080.

25                                        FIGURE 12 ABOUT HERE
                                          TABLE 6 ABOUT HERE

Future urban growth simulations were overlaid with flooding maps to show how different development patterns guided by distinct land use policies are vulnerable to sea level rise induced flooding (Tables 7 – 8). The results show that, if urban

growth progresses compactly, total inundated area in 2030 would be the largest among three growth scenarios. This finding is echoed by a previous study (Sekovski et al., 2015). In other words, compact development normally appears surrounding existing urban areas, the majority of which are low-lying and prone to flooding. With respect to land use polices, urban growth under regulations leads to less flood prone development. Specifically, if no regulations are implemented, on average





the flooded area in 2080 would be more than 25 times the area under flooding-risk mitigation. Both growth patterns and land use policies, accordingly, substantially affect the susceptibility of coastal cities to flooding hazards.

TABLE 7 – 8 ABOUT HERE

Figure 13 shows how three urban growth scenarios are exposed differentially to sea level rise induced flooding at a smaller geographical scale. First, urban growth is extremely limited if the excluded layer represents flooding-risk mitigation. Conversely, if water bodies are used as an excluded layer, urban areas expand considerably in coastlines and hinterlands. Second, three growth scenarios exhibit a similar spatial pattern regarding new urban areas. Third, a vast majority of

urbanized areas that would be within flooding polygons are situated in the proximity of the West and North Bay and the shoreline areas of Panama City. Flooded areas will increase apparently if no land use policy is applied, as shown in Figure 13 a, d, and g.

FIGURE 13 ABOUT HERE

**4.4 Discussion**

**4.4.1 Urban growth and coastal hazards**

SLEUTH calibration results indicate that main driving force for the study area is spread, followed by dispersion and breed. Such findings are consistent with similar coastal studies (Sakieh et al., 2015; Sekovski et al., 2015). In other words, urban growth is likely to take place around current settlements in a compact fashion. Existing settlements are featured by high

accessibility to infrastructure, activity centers, and coastal amenities. Additionally, a multitude of new urban areas cluster around coastlines, which is apparent either under historical growth or urban sprawl scenarios. Increased human activities and competition for limited resources, therefore, intensify environmental pressures and interest conflicts at the land-sea interface. Furthermore, the interface also faces unprecedented threat from sea level rise and other intensified coastal hazards. SLEUTH could provide useful information on future urban growth and thus benefit coastal city managers and land use planners.

However, such urban growth models have their own limitations. The most noticeable one is their inability of capturing the whole range of factors affecting urbanization (Herold, Goldstein, & Clarke, 2003). The demand for urban growth comes from population increase and economic development. The Bureau of Economic and Business Research at the University of Florida forecasts that the total population of Bay County will increase by almost 40% by 2030. Such an apparent rise probably demands significant urbanization. Therefore, socioeconomic factors behind urbanization should be considered in

applications. Population increase, however, is linked with population migration, overall economic conditions, and other factors, which is complicated and hard to predict. Additionally, urbanization in coastal regions is driven by economic



activities, the majority of which are related to tourism and real estate. These factors, nevertheless, are barely taken into account in CA models.

When urban growth models are coupled with coastal hazards that are associated with sea level rise, additional concerns arise. In SLEUTH urban growth predictions fail to incorporate seawall, population relocation, and other adaptation strategies to sea

level rise. This limitation can be seen in the given examples of future flooding risk (Figure 13). Considerable existing urban areas would fall into flooding polygons in 2030 and 2080. Essentially, the model assumes a "do-nothing" option with respect to adaptation strategies, which may be improved in future research. Another uncertainty arises during the development of sea level rise induced flooding. In other words, researchers have not yet reached an agreement as to sea level rise prediction. Sea level may possibly increase more rapidly than people initially thought. Nicholls and Cazenave (2010) reviewed numerous

prediction sources and concluded that global mean sea level would rise between 0.19 m and 1.7 m by 2100. Given these uncertainties, therefore, the simulation results should be interpreted with extreme carefulness and objectivity.

### 4.4.2 Policy implications

Compact urban forms are advocated because of their environmental friendliness and energy conservation (Dezhkam et al., 2013; A. Mahiny & Gholamalifard, 2007). Yet, this might not be true in coastal areas from the perspective of hazard

mitigation. As indicated in Tables 7 and 8, compact growth model generates more extension of current flood-prone areas than historical growth and urban sprawl. For instance, if flooding-risk mitigation is implemented, new urban areas that would be flooded in 2030 under compact growth pattern are over 2,500,000 $m^2$, almost double the area under historical growth scenario. While spontaneous urban growth should be regulated to prevent farmland loss, urban development should also be oriented towards hinterland which already has appreciable urban areas.

Building an integrated land use policy for urban growth landscape is a recommended option for coastal communities. The increased demand for urbanization and the goal for hazard mitigation can be coordinated through an integrative policy framework. In addition, land use zoning for existing coastal areas should be incorporated with adaptation strategies to sea level rise. Rural land use management and regulations should be oriented in order to attract new development inland. While the prohibition of development within flooding zones greatly constrains urban growth and is therefore unrealistic, only

relying on land use zoning could lead to considerable inundated urban areas. A compromise of these two alternatives, accordingly, could be developed to ensure adequate urbanization and steer new developments away from low-lying areas. Finally, seawall, planned retreat, and other adaptation strategies should be incorporated in the framework to protect existing urban areas from flooding risk. The modelling approach and results offered by this work could aid in the development of an integral land-use enforcement system.





## 5 Conclusions

Environmental and resource pressures are increasingly intensified given ongoing coastal urbanization. In addition, urbanization process amplifies the exposure of coastal communities to flooding hazards. Additionally, we are uncertain about the degree to which sea level rise may contribute to increased intensity of storminess. The possibility of the

exacerbated problem, however, cannot be neglected from a precautionary perspective. Therefore, building an effective coastal management plan to balance land use, competing interests, and hazard mitigation is crucial.

This work contributes to the literature by integrating urban growth dynamics, land use policies, and sea level rise induced flooding. We successfully calibrated the SLEUTH model for Bay County coastal areas, Florida, based on historical data from the year 1974 to 2013. By applying the best-fit coefficients we developed three urban growth scenarios and assessed

the exposure of future urban extent to sea level rise induced flooding under different land use policies. These scenarios reflected various growth strategies widely applied in urban planning. Our results indicate that urban growth is mainly driven by the parameter of compact development, and that substantial urban growth would be prone to coastal flooding if no land use policy is implemented.

SLEUTH is particularly useful in modelling complex spatial dynamics by applying the CA-based approach which captures

local interactions among cells. In addition, the computational capacity of SLEUTH is greatly enhanced due to rapid advancement of computer technologies. By outputting GIF maps and statistics for each prediction year, SLEUTH can be easily linked with a raster-based GIS environment (Rafiee et al., 2009). Therefore, the results of different scenarios can be readily imported into a GIS platform for presentation purposes. Coupled with hazards maps, the model serves as a decision support tool and helps city managers, land use planners, and hazard mitigation teams evaluate the outcomes of different

policies. Additionally, the visualization results can be used to raise general awareness about the vulnerability of coastal communities to sea level rise.

As mentioned, the model is just a simplification of reality, and urban growth is an intricate process which involves population increase, economic activities, and many other factors. Since the level of impact of sea level rise on flooding is quite unclear, and growth predictions could be probably biased, the model cannot generate exact results of urban growth and

flooding extent. However, we believe that "what-if" estimations are useful in helping decision makers understand how different developments may be oriented. Probabilistic models and scenario-based planning, therefore, should be advocated in order to evaluate planning alternatives and their consequences (Xiang & Clarke, 2003) as well as offer reliable estimates of flooding damages.

*Acknowledgements*

The authors thank Dr. Keith C. Clarke for his help on the model calibration process. The authors also thank the U.S. Census Bureau, the Florida Geographic Data Library, the Bay County Online, and the Bureau of Economic and Business Research for offering access to data. This paper was undertaken with support from the Florida Sea Grant, Grant No. R/GOM-RP-2, "A





Parameterized Climate Change Projection Model for Hurricane Flooding, Wave Action, Economic Damages, and Population Dynamics". This paper was also funded by the Florida Sea Grant Project entitled "A Spatial-Temporal Econometric Model to Estimate Costs and Benefits of Sea-Level Rise Adaptation Strategies".

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

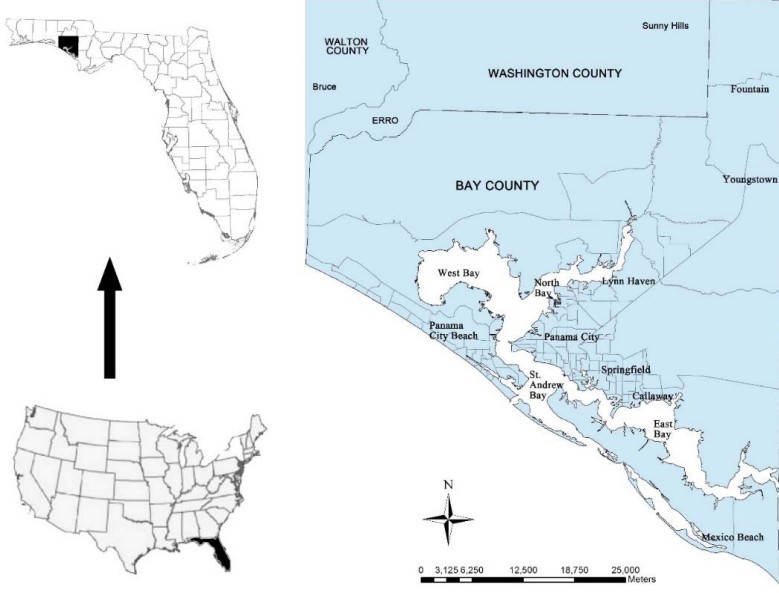

**Figure 1: Study area**





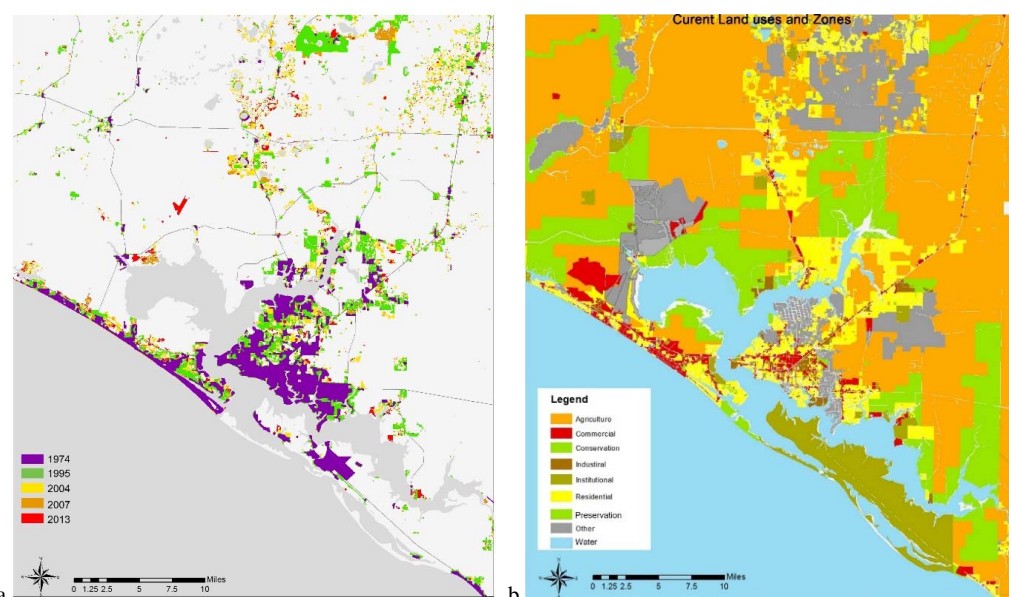

**Figure 2: Historical urban changes (a) and current land uses and zones (b) for the study area**

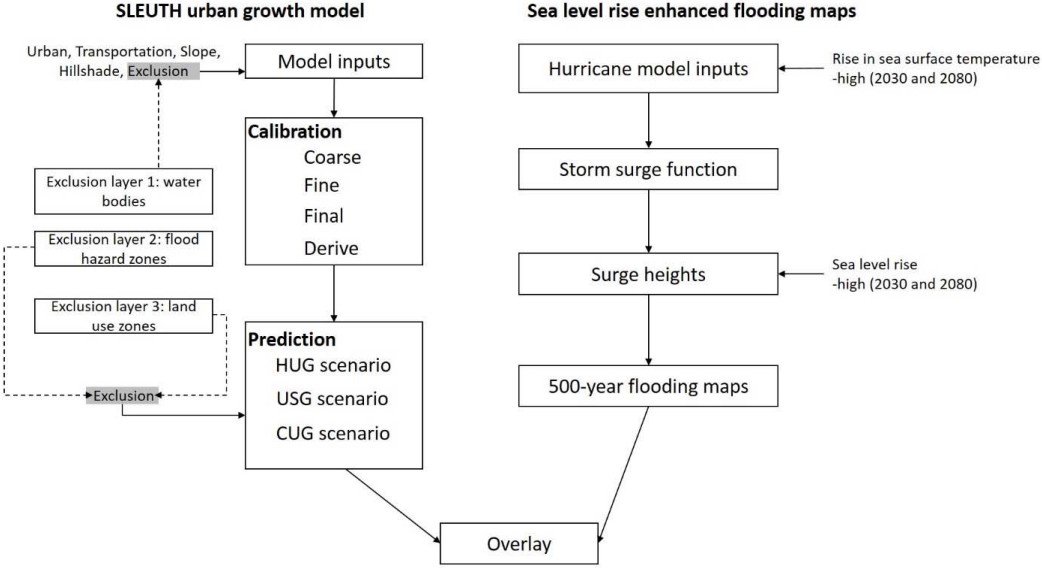

5    **Figure 3: Overall study framework**




**Figure 4: Urban layers, transportation, topographic and historic excluded layer for model inputs (100 is the completely exclude area)**





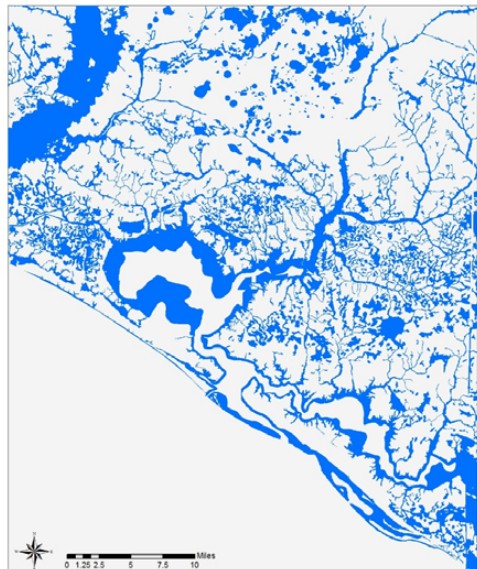

**Figure 5: Special Flood Hazard Area in 1996**

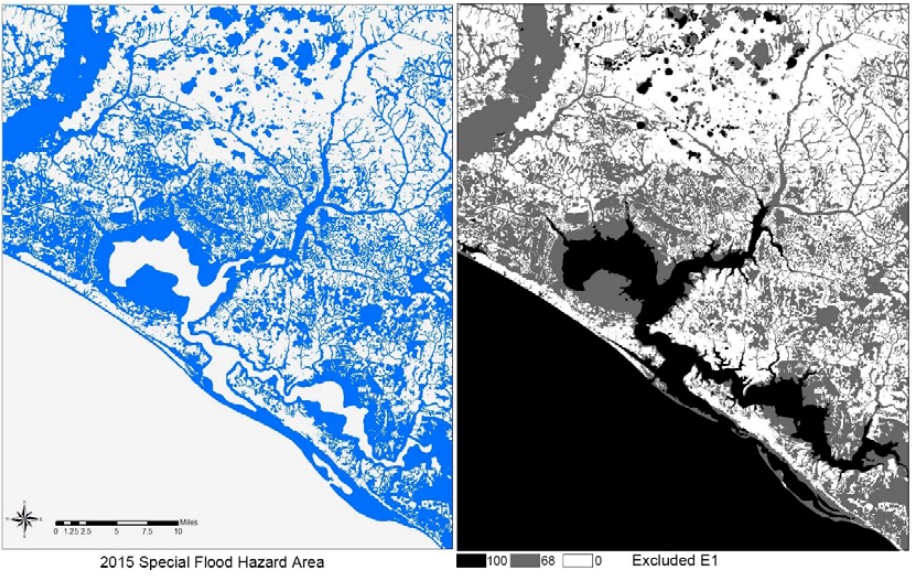

5    **Figure 6: E1 excluded layer (100 is the entirely excluded area)**





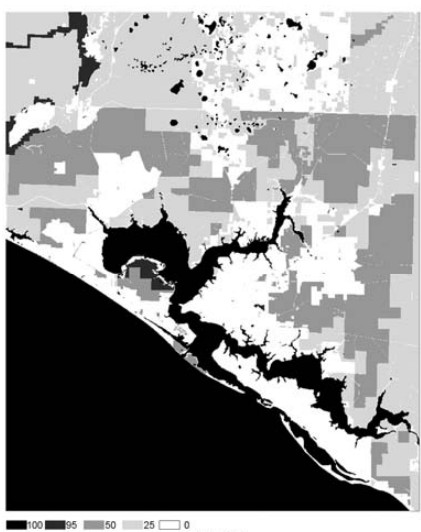

**Figure 7: E2 excluded layer (100 is the entirely excluded area)**

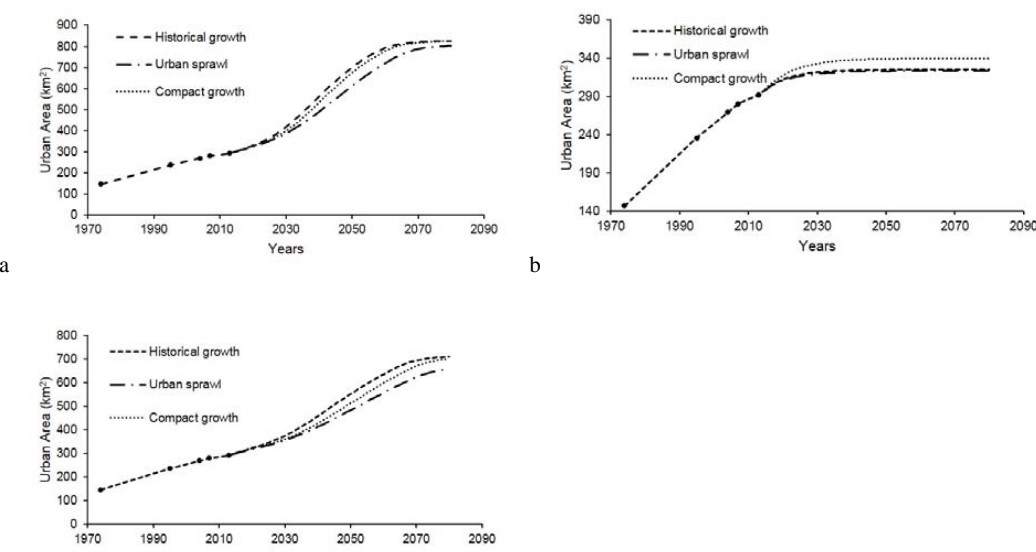

a                                                     b

5    c

**Figure 8: Simulation of urban changes to the year 2080 under three scenario for different excluded layers. a) E0-water bodies, b) E1-flooding-risk mitigation, c) conservational/agricultural land protection**




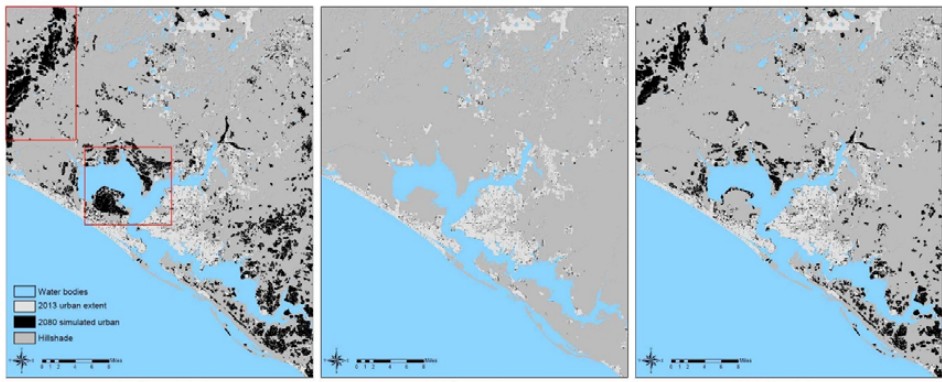

**Figure 9: The comparison of modelled urban extent in 2080 under historical growth predicted with different excluded layers**

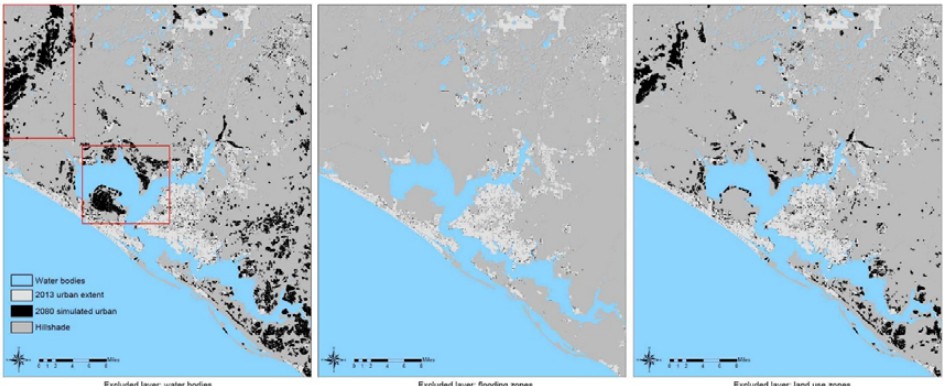

**Figure 10: The comparison of modelled urban extent in 2080 under urban sprawl predicted with different excluded layers**

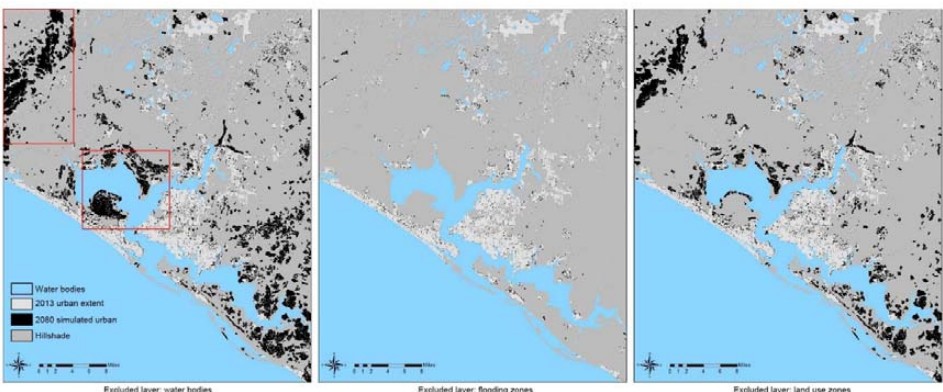

**Figure 11: The comparison of modelled urban extent in 2080 under compact development predicted with different excluded layers**



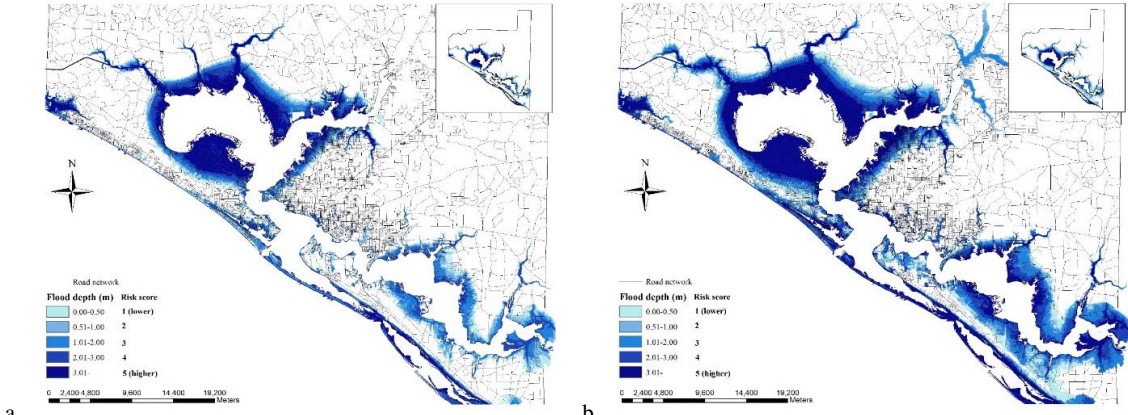

a    b

**Figure 12: 500-year flood-risk zones of 0.2-m sea level rise (SLR) (a) and 0.9-m SLR (b)**





5 **Figure 13: The comparison of 500-year flooding with the HUG (a-c), USG (d-e), and CUG (f-i) up to 2080 under three excluded layers**



**Table 1. Sources and descriptions of data sets**

| Data type | Year | Spatial resolution | Description and source |
|---|---|---|---|
| Urban extent | 1974 | Vector files | Land use/cover maps from Florida Geographic Data Library |
| | 1995 | | |
| | 2004 | | |
| | 2007 | | |
| | 2013 | | |
| Transportation network | 2007 | Vector files | TIGER/Line® Shapefiles and TIGER/Line® Files |
| | 2013 | | |
| Slope | - | 30 * 30 m | Converted from National Elevation Datasets of Geospatial Data Gateway from U.S. Natural Resources Conservation Service |
| Hillshade | - | | |
| Exclusion | 1996 | Vector files | Flood Insurance Rate Maps from Florida Geographic Data Library |
| | 2015 | | |
| | 2016 | Vector files | Land uses and zones from Bay County GIS online |
| Flooding maps | 2030 | - | The estimates of sea level rise and sea surface temperature published by the IPCC |
| | 2080 | | |

**Table 2. The relationships among a growth cycle, transitional steps, and controlling factors (Clarke et al., 1997)**

| Growth cycle | Transitional steps | Controlling factors | Description |
|---|---|---|---|
| 1-1 | Spontaneous growth | Dispersion | Random urbanization of cells |
| 1-2 | New spreading centers | Breed | Outward growth of new settlements formed in the spontaneous growth stage |
| 1-3 | Edge growth | Spread and slope | Outward growth of current settlements |
| 1-4 | Road-influenced growth | Road gravity, dispersion, breed, and slope | Growth of new settlements within a distance of existing transportation networks |

5    **Table 3. The development of excluded value for E1 scenario**

| Zones | New growth from 1995 to 2013 (hectares) | Total area (hectares) | Growth rate (‰) | Excluded value |
|---|---|---|---|---|
| Special Flood Hazard Area | 665 | 92,054 | 0.4 | 68 |
| Non-SFHA | 4,901 | 216,095 | 1.3 | 0 |
| Water bodies | - | - | - | 100 |

**Table 4. SST Changes and SLR in Two Future Years in Panama City, FL**

| Climate Scenario | Rise in SST (℃) | SLR (m) |
|---|---|---|
| Present-day | 0 | 0 |
| A1FI (high) in 2030 | 1.23 | 0.20 |
| A1FI (high) in 2080 | 5.02 | 0.90 |





**Table 5. Growth parameters for three scenarios**

| Scenarios | Dispersion | Breed | Spread | Slope | Road gravity |
|---|---|---|---|---|---|
| Historical growth | 71 | 92 | 70 | 3 | 35 |
| Urban sprawl | 96 | 100 | 60 | 3 | 55 |
| Compact development | 46 | 77 | 80 | 3 | 10 |

**Table 6. Flooded areas under 0.2-m and 0.9-m sea level rise scenarios**

| Flooding depth (m) | Area (km$^2$) | |
|---|---|---|
| | 0.2-m sea level rise (2030) | 0.9-m sea level rise (2080) |
| 0 – 0.5 | 4.6 | 58.8 |
| 0.5 – 1 | 4.3 | 52.5 |
| 1 – 2 | 7.0 | 100.3 |
| 2 – 3 | 7.0 | 77.7 |
| Greater than 3 | 8.2 | 176.1 |
| Total | 31.1 | 465.4 |

5    **Table 7. The results of overlaying urban growth predictions with the 500-year flooding map in 2030**

| | E0: no regulations | | E1: flooding-risk mitigation | | E2: conservational/agricultural land protection | |
|---|---|---|---|---|---|---|
| | Total flooded area (m$^2$) | New urban areas that would be flooded in 2030 (m$^2$) | Total flooded area (m$^2$) | New urban areas that would be flooded in 2030 (m$^2$) | Total flooded area (m$^2$) | New urban areas that would be flooded in 2030 (m$^2$) |
| HUG | 59,112,900 | 7,254,000 | 53,223,300 | 1,364,400 | 58,479,300 | 6,620,400 |
| USG | 57,466,800 | 5,607,900 | 52,974,900 | 1,116,000 | 56,867,400 | 5,008,500 |
| CUG | 59,185,800 | 7,326,900 | 54,468,900 | 2,610,000 | 58,603,500 | 6,744,600 |

**Table 8. The results of overlaying urban growth predictions with the 500-year flooding map in 2080**

| | E0: no regulations | | E1: flooding-risk mitigation | | E2: conservational/agricultural land protection | |
|---|---|---|---|---|---|---|
| | Total flooded area (m$^2$) | New urban areas that would be flooded in 2080 (m$^2$) | Total flooded area (m$^2$) | New urban areas that would be flooded in 2080 (m$^2$) | Total flooded area (m$^2$) | New urban areas that would be flooded in 2080 (m$^2$) |
| HUG | 200,035,800 | 116,387,100 | 87,068,700 | 3,420,000 | 168,683,400 | 85,034,700 |
| USG | 185,077,800 | 101,429,100 | 86,526,900 | 2,878,200 | 141,254,100 | 57,605,400 |
| CUG | 199,561,500 | 115,912,800 | 89,570,700 | 5,922,000 | 164,392,200 | 80,743,500 |