# Peer review of "Published: 13 July 2016"

_Natural Hazards and Earth System Sciences, 2016_

## Referee Comment (RC1) · Anonymous Referee #1 · 6 Sep 2016

The authors have addressed an important issue in Land Change Science by addressing multiple issues relevant to the field simultaneously. First, they employed a CA model to understand historical parameters of land use change in the area of interest. Second, they used this information to explore future scenarios of generalized land use policy cross-dimensionally with different levels of SLR. The issue of managing SLR adaptation in low-lying areas is absolutely critical and so this research is of great urgency.

Having said that, I do have some comments on the methods used to reach the conclusions cited at the end of this paper. First, the authors borrow from the techniques developed by Onsted and Roy Chowdhury (2014) but they missed one of the most important points of that paper: measuring urban growth over an entire zone is not an accurate way of understanding differential impacts over a heterogeneous landscape.

[Figure]

The results of their work show that utilizing such a technique results in worse goodness of fit metrics than treating the landscape as homogeneous. Instead, the authors recommend using the AMLEG technique which helps to address distance decay effects that dilute the efficacy of measuring all urban growth in a large zone. I strongly suggest the authors take a look at this technique (in Onsted and Roy Chowdhury (2014)) as its employment will increase the accuracy of the E values in the E-1 scenario.

Second, the authors made no attempt to scientifically derive the differences amongst the land use zoning categories in their construction of E2 as they did for E1. Instead, they just guessed because others have done so. However, other results from Onsted and Roy Chowdhury (2014) suggest that guessing results in poor accuracy as well, or at least worse than treating the entire area as homogeneous.

Third, best practice usually discourages forecasting further into the future than you have calibrated in the past. The authors have data from 1974 to 2013, which is 39 years of data. But they use this to forecast 67 years into the future. Please see the figure from Goldstein et al. (2014).

Goldstein, N.C., J.T. Candau, and K.C. Clarke. 2004. "Approaches to simulating the "March of Bricks And Mortar"". Computers, Environment and Urban Systems 28:125-147.

Fourth, the future scenario results are less emergent from interacting and hard to predict factors than they were engineered by the authors to fit a priori expectations. For example, a sprawl scenario was designed by tweaking the model's growth parameters until sprawl was achieved. However, it is important to reflect upon the utility of such scenarios for understanding the nature of urban growth in the area as well as how it can help us improve our modeling methods. For example, from a policy maker point of view, what suite of policies will lead to such a sprawl scenario? What suite of policies will lead to a compact growth scenario? The sprawl and compact growth scenarios are implicit in the sense that they suggest "if a series of circumstances happen that result

in sprawl, this is what sprawl would look like". However, do we need a model to tell us what future sprawl will look like if we already decide what that future is? The more the outcome is controlled the less predictive quality the model has.

I suggest that authors, at most, redo the aspects of the methods described above. However, at the least, the issues I mention above should be discussed in the article as possible deficiencies in the current methodology.

The strongest aspect of this work is the integration of SLR scenarios vis-à-vis future urban growth scenarios. This line of inquiry is absolutely critical for coastal resilience and thus the authors should be applauded for the great importance this kind of research has on the sustainability of our worldwide coastal civilizations.

Minor Comments:

Page 4, Lines 25 thru 30: The impervious surface percentages for the various zoning categories mentioned are incorrect (I checked). The authors need to revise this accordingly.

Page 6, Line 12: This should read "percentage slope" instead of percentage rise since percent slope is rise over run.

Page 10, Line 14: Increased SST does not cause higher pressure over the ocean, but lower pressure. The magnitude of a hurricane is often directly related to how LOW the pressure in the eye is, thus Lower is stronger. The authors seem to have this reversed.

Page 11, Lines 12 thru 18: The methods the authors list (differences of 25, etc.) does not match what they actually have listed in Table 5.

Page 14, Line 14: The authors seem to suggest that sprawl leads to less vulnerability in all coastal areas and thus policy makers must choose as a tradeoff between sprawl with all of its negative environmental consequences or flooding. However, the most important factor is not necessarily proximity to the coast but, rather, simple elevation. Thus Panama City is not necessarily representative of the topographical constraints

and opportunities in all coastal areas.

Page 14, Lines 28 – 29: As Florida sits on porous limestone a seawall will not keep out SLR since the ocean will just come up underneath on the other side.

Figure 1: There are three maps at three different scales but only one scale bar is used. Each frame should have its own scale bar.

Figure 6: Though technically an Excluded Layer can be portrayed however one wants in the actual publication it is confusing to see the actual Excluded Layer in grayscale but portraying the opposite grayscale values of their E scores. Thus the ocean should be 100 or over, but instead is represented as 0, etc. It could be helpful to those in the SLEUTH community if the authors showed the grayscale Gif Excluded layers exactly as they are.

Figure 9A, 10A, and 11 A: The bounding boxes should be removed as they serve no purpose. They should instead be used in Figure 12 as it appears that is where they correspond.

Figure 13: The dark blue color appearing in the maps of this Figure does not appear in the legend. I am assuming it is urbanized land that is also flooded. However, guesswork should not be required by the reader.

Technical: There are numerous spelling issues, missing articles, etc. throughout the manuscript. For example, "Talbe 4 About here". Or poses "unnecessarily" risk instead of poses "unnecessary" risk in the abstract. Another pass of proofreading is recommended.
* * *

---

## Author Comment (AC1) · 4 Oct 2016

Dear Reviewer,

The authors are highly grateful for the insightful comments and suggestions. Based on the comments, the manuscript was revised and improved. The authors meticulously reviewed and proofread the manuscript to ensure that the revisions addressed every aspect of the comments. Below please find our detailed responses to the comments. Again, thank you very much for your time and efforts on reviewing our paper.

General comments:

1. The authors have addressed an important issue in Land Change Science by addressing multiple issues relevant to the field simultaneously. First, they employed a

CA model to understand historical parameters of land use change in the area of interest. Second, they used this information to explore future scenarios of generalized land use policy cross-dimensionally with different levels of SLR. The issue of managing SLR adaptation in low-lying areas is absolutely critical and so this research is of great urgency.

Response: We totally agree with the reviewer. Hurricane Katrina, Hurricane Sandy, and other disastrous coastal hazards have led to enormous economic losses and serious life consequences. These lessons have urged coastal communities to take actions to protect their residents from coastal hazards. Thus, the research on SLR adaptation is of great importance. Additionally, it is crucial to identify potentially vulnerable areas to SLR, and therefore local governments can efficiently allocate recourses to optimize the deployment of SLR adaptation strategies. This study was thus initiated in order to develop an integrated framework that connects urban dynamics and SLR related hazards.

2. Having said that, I do have some comments on the methods used to reach the conclusions cited at the end of this paper. First, the authors borrow from the techniques developed by Onsted and Roy Chowdhury (2014) but they missed one of the most important points of that paper: measuring urban growth over an entire zone is not an accurate way of understanding differential impacts over a heterogeneous landscape. The results of their work show that utilizing such a technique results in worse goodness of fit metrics than treating the landscape as homogeneous. Instead, the authors recommend using the AMLEG technique which helps to address distance decay effects that dilute the efficacy of measuring all urban growth in a large zone. I strongly suggest the authors take a look at this technique (in Onsted and Roy Chowdhury (2014)) as its employment will increase the accuracy of the E values in the E-1 scenario.

Response: Thank you for the comments. We acknowledge that urban growth over a large study area is heterogeneous. Therefore, we developed a new E3 excluded layer that was based on the AMLEG approach. Significant improvements have been made in

terms of overall research framework, the methodology, results, and discussions. First, the new E3 excluded layer was incorporated into the new study framework (Figure 3 on Page 7). Second, a new section was added to the methodology part to illustrate the procedures by which the flooding-risk mitigation based the AMLEG was developed. In section 3.2.4, we will add "Most SLEUTH modelers apply the above-mentioned methods to develop excluded layers. However, such methods are deficient since they usually treat the whole study area homogeneously. Conversely, urban growth is more likely to involve heterogeneous changes across the study area. For instance, in coastal regions new residential developments largely extend from existing settlements that may only cover a small portion of the whole study region. Thus, Onsted and Chowdhury (2014) developed a procedure that corrects the growth rates in the AMLEG. The authors concluded that the AMLEG technique produced more accurate results than the other two methods: arbitrary guessing and the calculation based on the whole study area. Therefore, the AMLEG was applied based on the E1 scenario (flooding mitigation). The SLEUTH was run in the prediction mode for 100 Monte Carlo times for the period of 1995 to 2013. All five growth coefficients were set as 100, and the cells with an urbanization probability of 50% or more were considered in the AMELG, as shown in Figure 7" [Page 12, Line 1 through Page 13, Line 5].

In the section of results and discussions, we made the following improvements.

1) Page 16, Line 30 through Page 17, Line 5: The quantity of urban growth under the E3 excluded layer was discussed and compared with the other policy scenarios.

2) Page 17, Line 8 through Line 12: Urban growth up to 2080 under the E3 scenario was illustrated in Figure 10 and discussed in the text.

3) Page 19, Line 3 through Page 20, Line 4: The AMLEG effects on urban exposure to sea level rise induced flooding were shown in Tables 8-9 and Figure 12 and analyzed in the discussions.

3. Second, the authors made no attempt to scientifically derive the differences amongst

the land use zoning categories in their construction of E2 as they did for E1. Instead, they just guessed because others have done so. However, other results from Onsted and Roy Chowdhury (2014) suggest that guessing results in poor accuracy as well, or at least worse than treating the entire area as homogeneous.

Response: The authors appreciate the comments. This is a limitation of our work. Although we tried to cite reliable sources to develop adequately scientific estimates for the excluded values of E2, we fail to point out potential bias due to the experience-based method in the original manuscript. The E2 layer should be recreated using historical data and more robust techniques such as the AMLEG, but we had difficulty collecting historical zoning maps corresponding to the past land cover maps. Therefore, we will highlight and discuss the methodological deficiencies by making the following improvement in the revised manuscript.

Page 22, Line 19 through Line 22: we will add "Third, urban growth predictions may be biased if modelers fail to justify the values in excluded layers. This study determined the excluded values of E2 scenario according to future land use plans and suggestions from other studies (e.g., Akin et al., 2014). Although the lack of historical zoning information forced us to make this assumption, the predictions under the E2 scenario may become problematic".

4. Third, best practice usually discourages forecasting further into the future than you have calibrated in the past. The authors have data from 1974 to 2013, which is 39 years of data. But they use this to forecast 67 years into the future. Please see the figure from Goldstein et al. (2014). Goldstein, N.C., J.T. Candau, and K.C. Clarke. 2004. "Approaches to simulating the "March of Bricks And Mortar"". Computers, Environment and Urban Systems 28:125-147.

Response: Thank you for the insightful comments. The authors totally agree that urban growth predictions should not exceed the time range of calibration. In initial experiments, we considered a short time frame but found that sea level rise impact

was extremely limited. Thus, the authors used a longer forecasting time range to make significant the sea level rise consequences. In addition, some studies may suggest that scenario-based predictions may choose an aggressive extrapolation option to show any emerging patterns, while the interpretation of results should be very careful. This can be seen in a recent report that forecasts future urban developments in 2070 for the whole Florida (Mapping Florida's Future – Alternative Patterns of Development in 2070, retrieved through: http://1000friendsofflorida.org/florida2070/wp-content/uploads/2016/09/florida2070technicalreportfinal.pdf). However, the authors made the following discussions regarding the issues associated with extrapolation.

Page 22, Line 30 through Page 23, Line 4: we will add "Given these uncertainties, therefore, the simulation results should be interpreted with extreme carefulness and objectivity. In addition, the extrapolation of future urban growth beyond the calibration range can be questionable and generate uncertain results (Goldstein, Candau, & Clarke, 2004). Modelers ought to make a trade-off between urban predictions and the forecasts of climate change related hazards. Climate change is slow going, but urbanization may be rapid in populated coastal regions. For instance, SLR may become significant only after an adequate time frame that probably exceeds the time period of historical urban data. Such coupled analyses should aim at identifying the general impacts of climate change on prospective urbanization, rather than replicating past patterns of urban development".

5. Fourth, the future scenario results are less emergent from interacting and hard to predict factors than they were engineered by the authors to fit a priori expectations. For example, a sprawl scenario was designed by tweaking the model's growth parameters until sprawl was achieved. However, it is important to reflect upon the utility of such scenarios for understanding the nature of urban growth in the area as well as how it can help us improve our modeling methods. For example, from a policy maker point of view, what suite of policies will lead to such a sprawl scenario? What suite of policies will lead to a compact growth scenario? The sprawl and compact growth scenarios are

implicit in the sense that they suggest "if a series of circumstances happen that result in sprawl, this is what sprawl would look like". However, do we need a model to tell us what future sprawl will look like if we already decide what that future is? The more the outcome is controlled the less predictive quality the model has.

Response: The authors are very grateful for the comments. Many historical urban growth cases have witnessed urban sprawl and compact development. These urban forms are two of the more representative urban growth patterns than others. So we choose them to examine the sea level rise impacts on possible urban change scenarios. In fact, scenario-based predictions were adopted by numerous similar studies (see, for example, Refiee et al. (2009), Zheng et al. (2015), and Sakieh et al. (2015)). We admit that more model controls mean less flexibility, so we used the historical growth pattern as a reference. Additionally, we highly agree with the referee that it is crucial to identify the policy implications of various growth patterns, which is absent in many simulation papers. Thus, we reorganized the section of policy implications by following a logically connected flow of the text. First, we discussed how urban forms were exposed to sea level rise and which urban form appeared less vulnerable to climate change related hazards. Following this, we expanded the discussions by talking about other dimensions of urban forms such as their influence on social-ecological welfare and different policies leading to different urban patterns. Lastly, we narrowed down our discussions by proposing general policy frameworks that promote both societal prosperity and hazard mitigation. In summary, in the new manuscript, we made the following changes.

Page 23, Line 16 through Line 29: we reviewed policies leading to urban sprawl. "Population and economic growth largely drive urban growth. Yet, policies behind such growth should also be investigated since land use plans and economic strategies represent developmental blueprints. Thus, it is beneficial to reflect upon different policies contributing to distinct urban growth patterns: urban sprawl and compact growth. Urban sprawl is characterized by unplanned and scattered developments in suburban areas. Uncoordinated growth in the city edge has been suggested to be associated with multidimensional factors regarding economic incentives, housing development plans, and transportation policies (De Vos & Witlox, 2013; Lopez & Hynes, 2003; Yue, Zhang, & Liu, 2016). Economic incentive packages launched by the central government have contributed to urban sprawl in the developing world. For instance, China took economic reform in the 1970s by opening up land markets and commercializing housing units. This economic stimulus gave rise to many sprawling mega-cities such as Beijing, Shanghai, and Chengdu. In Europe and North America, micro-economic theories may explain urban sprawl. Households begin to relocate to the suburbs when the land prices in city centers become prohibitively high. Their relocation decisions are further strengthened by housing and land development policies. Developers promote low-density communities in the city periphery. Local governments help to build large retail centers to accommodate the increased demand. Motorization policies and low fuel costs result in automobile-oriented cities. Even public transit policies aggravate outward city growth by charging long-distance commuters less than riders for short distances (De Vos & Witlox, 2013)".

Page 23, Line 29 through Page 24, Line 6: The policies promoting compact development were reviewed. "As urban sprawl increasingly threatens public health, social equity, and the built environments, people start to develop different urban containment policies. There are primarily two forms of containment policies that were adopted in the US. The State law in Oregon and Washington requires that local land-use plans should clearly define an urban growth boundary. In other states such as Florida and Maryland, governments develop urban service limits, public facilities ordinances, and other policies to promote compact urban forms (Aytur, Rodriguez, Evenson, & Catellier, 2008). However, the effects of urban containment policies have been hotly debated. For example, not all urban growth boundaries significantly affect housing markets and urbanization paces (Dempsey & Plantinga, 2013). Thus, De Vos and Witlox (2013) suggest the integration of spatial planning policies, mobility policies, and road pricing. Spatial planning can strictly limit new developments outside urban areas. The development of Transit-Oriented Development benefits nurturing high-density and mixed land-use neighborhoods. In addition, road pricing increases long-distance travel costs, thereby curtailing urban sprawl".

6. I suggest that authors, at most, redo the aspects of the methods described above. However, at the least, the issues I mention above should be discussed in the article as possible deficiencies in the current methodology.

Response: Thank you for the suggestions. We have spent great efforts on redoing the crucial aspects regarding the current methodology. As stated in the response to the second comment, a new E3 excluded layer was created based on the AMLEG technique, and some parts of the simulation were run again to incorporate this change. Accordingly, substantial changes were made in terms of the methodology, the visualization of results, and discussions. However, we fail to redo a few methodological aspects raised by the referee due to data limitations. We did highlight and discuss these method limitations in the discussion section [see Page 22, Line 1 through Page 24, Line 17].

7. The strongest aspect of this work is the integration of SLR scenarios vis-à-vis future urban growth scenarios. This line of inquiry is absolutely critical for coastal resilience and thus the authors should be applauded for the great importance this kind of research has on the sustainability of our worldwide coastal civilizations.

Response: We are grateful for the referee's applause. We hope that the research could be a small yet important effort in the coastal resilience science. Informative decision support tools can greatly benefit coastal communities and mitigate the economic and life consequences of coastal hazards.

Specific comments:

1. Page 4, Lines 25 thru 30: The impervious surface percentages for the various zoning categories mentioned are incorrect (I checked). The authors need to revise this

accordingly.

Response: The authors apologize for the confusion. We carefully read the source document again and rephrased the explanations to make clear the impervious surface percentages [see Page 5, Line 19 through Page 6, Line 6].

2. Page 6, Line 12: This should read "percentage slope" instead of percentage rise since percent slope is rise over run.

Response: Thank you for pointing out the accurate terminology. The text was then corrected according to this comment [Page 8, Line 3].

3. Page 10, Line 14: Increased SST does not cause higher pressure over the ocean, but lower pressure. The magnitude of a hurricane is often directly related to how LOW the pressure in the eye is, thus Lower is stronger. The authors seem to have this reversed.

Response: Thank you for this comment. The authors carefully went through the original citation and found that we had some misunderstanding on this phenomenon. Thus, the text was revised based on this comment [Page 15, Line 9].

4. Page 11, Lines 12 thru 18: The methods the authors list (differences of 25, etc.) does not match what they actually have listed in Table 5.

Response: Thank you for the correction. We found the typo in Table 5 and revised it accordingly [Page 16, Line 15].

5. Page 14, Line 14: The authors seem to suggest that sprawl leads to less vulnerability in all coastal areas and thus policy makers must choose as a tradeoff between sprawl with all of its negative environmental consequences or flooding. However, the most important factor is not necessarily proximity to the coast but, rather, simple elevation. Thus Panama City is not necessarily representative of the topographical constraints and opportunities in all coastal areas.

Response: We appreciate the comment. The previous statement suffered from the inappropriate generalization of the exposure of different urban forms to sea level rise. Thus, we rephrased the statement and the conclusion about urban vulnerability. We will add: "Yet, this might not be true in flat coastal areas from the perspective of hazard mitigation [Page 23, Line 7] ... however, such conclusions are made only based on our case study area where the elevation is low and change insignificantly[Page 23, Line 11]".

6. Page 14, Lines 28 – 29: As Florida sits on porous limestone a seawall will not keep out SLR since the ocean will just come up underneath on the other side.

Response: Thank you for the comments. The following text was added to include this information suggested by the referee. "In the long term, adaptation plans ought to consider other SLR aspects such as groundwater pollution and saltwater intrusion beneath protective structures [Page 24, Line 15]".

7. Figure 1: There are three maps at three different scales but only one scale bar is used. Each frame should have its own scale bar.

Response: Thank you for the comments. The figure was improved with each map having its own scale bar. The enhanced figure could be found on page 4.

8. Figure 6: Though technically an Excluded Layer can be portrayed however one wants in the actual publication it is confusing to see the actual Excluded Layer in grayscale but portraying the opposite grayscale values of their E scores. Thus the ocean should be 100 or over, but instead is represented as 0, etc. It could be helpful to those in the SLEUTH community if the authors showed the grayscale Gif Excluded layers exactly as they are.

Response: We appreciate the comments. The authors apologize for such confusions. Thus, in our new figures, the water bodies were not assigned any excluded values. We hope this improvement helps to clearly depict the excluded layers [see Figure 4 on

Page 9, Figure 5 and 6 on Page 11, and Figure 8 on Page 13].

9. Figure 9A, 10A, and 11 A: The bounding boxes should be removed as they serve no purpose. They should instead be used in Figure 12 as it appears that is where they correspond.

Response: We removed the bounding boxes according to the referee's suggestion. In addition, these three figures were combined to reduce the total number of figures [Figure 10 on Page 18].

10. Figure 13: The dark blue color appearing in the maps of this Figure does not appear in the legend. I am assuming it is urbanized land that is also flooded. However, guesswork should not be required by the reader.

Response: Thank you for the comment. A complete legend was added in Figure 12. In addition, the authors greatly enhanced the readability and appearance of figures by increasing figure resolution, enlarging legends and texts, and redesigning the layout [see Figure 2, Figure 4, Figure 7, Figure 9, Figure 10, and Figure 12].

11. Technical: There are numerous spelling issues, missing articles, etc. throughout the manuscript. For example, "Talbe 4 About here". Or poses "unnecessarily" risk instead of poses "unnecessary" risk in the abstract. Another pass of proofreading is recommended.

Response: The authors highly apologize for grammatical mistakes. We have carefully proofread the whole manuscript to exclude language issues as much as possible. Special attention has been given to misspelled words and article issues. Furthermore, we inserted figures and tables into the main text and rearrange the overall structure in the new manuscript. Readers can then relate the information with illustrations and numbers more easily.

Please also note the supplement to this comment:

http://www.nat-hazards-earth-syst-sci-discuss.net/nhess-2016-157/nhess-2016-157-AC1-supplement.pdf

**Supplement:**

**An examination of land use impacts of sea level rise induced flooding**

Jie Song[1], Xinyu Fu[1], Yue Gu[1], Yujun Deng[1], Zhong-Ren Peng[1]

[1]Department of Urban and Regional Planning, College of Design, Construction, and Planning, University of Florida, POB 115706, 32611, USA

5  *Correspondence to*: Zhong-Ren Peng (zpeng@ufl.edu)

**Abstract.** Coastal regions are under intense developments because of their biodiversity and economic attractiveness. Meanwhile, these places are highly vulnerable to coastal hazards that are associated with sea level rise. Continuing urban development in these coastal areas that are prone to flooding increasingly poses unnecessary risk to their residents. While overwhelming efforts have been made to investigate coastal land use changes, few studies have simultaneously explored the

10  urban growth dynamics and its interaction with coastal hazards. This paper applied the cellular automaton-based SLEUTH model to calibrate historical urban growth patterns from 1974 to 2013 in Bay County, Florida, USA. Three scenarios of urban growth---historical trend, compact development, and urban sprawl---up to 2080 were predicted by applying the calibrated SLEUTH model. To assess the effects of different policies, we developed four excluded layers: no regulations, flooding-risk mitigation based on the whole study region, conservational/agricultural land protection, and flooding-risk mitigation under the

15  scenario of Areas More Likely to Experience Growth. We then evaluated how different urban growth patterns were oriented under these policies. Eventually, flooding maps were overlaid with future urban areas, and the exposure of different urban growth patterns to sea level rise induced flooding was examined. The findings suggest that if the coastal cities expand in a compact manner, areas vulnerable to flooding will increase compared with historical trend and urban sprawl scenarios. With respect to policies, if no regulations are implemented, on average the flooded area in 2080 would be more than 25 times under

20  flooding-risk mitigation. The joint model can serve as a decision support tool to assist city officials, urban planners, and hazard mitigation teams in making informed decisions. The visualization results can be useful in public outreach regarding coastal communities' increasing risk to sea level rise induced flooding.

**1 Introduction**

[revised manuscript text omitted]

SLEUTH applications focused on assessing prospective urban sprawl, compact development, and other growth patterns, the investigation of future land use exposure to coastal flooding received less attention. To the best of our knowledge, only two studies (Garcia & Loáiciga, 2014; Sekovski et al., 2015) attempted to couple land use predictions with marine flooding maps using SLEUTH. The combination of SLEUTH simulations and flooding hazard maps, however, should be prioritized in coastal land use modelling to better inform spatial management (Jeffrey A. Onsted & Chowdhury, 2014). Therefore, this study aims to evaluate the extent to which different urban growth patterns may be exposed to SLR induced flooding. Specifically, two research questions of this study are:

1. How may different urban growth patterns increase regional vulnerability to SLR induced flooding?
2. Would land use zoning and flooding mitigation plans help to steer prospective developments away from flood-prone regions?

This paper is organized as follows. Section 2 describes the study area and data collection. Following this, section 3 illustrates the modelling framework and outlines major steps for calibration, prediction, and the floodplain generation. In section 4 we present the calibration coefficients and discuss forecasting outcomes that were overlaid with flooding maps. Finally, section 5 offers a brief conclusion and provides the outlook for future research.

**2 Study area and data description**

**2.1 Study area**

The study region is located in northwest Florida and has long shorelines which form four bays along the Gulf Coast (Figure 1). It covers Bay County and some areas of Washington and Walton County. Its topographical features and past land use development render this region extremely susceptible to storm surges and hurricanes. For instance, considerable residential and commercial buildings encroached upon seafront areas in Panama City due to the absence of land use regulations in the past; as a result, urban growth largely occurs in flood-risk zones (Bay County Online, 2016). Historically, the study area has been hit by seventeen hurricanes since 1877 (Hurricanecity, 2015). Among these incidences, Hurricane Eloise in 1975 led to $23.1 million damages in structures, seawalls, and patios (Shows, 1978).

The economic activities and population profile of the study area increase its vulnerability to coastal hazards as well. Bay County highly relies on tourism related industries; specifically, restaurant and real estate are major industry sectors in Panama City. In Bay County the total spending on tourism as of June 2015, which was $121 million, had doubled since 2008 (Bureau of Economic and Business Research, 2015). Bay County has 10,222 firms, of which approximate 90% are small businesses. Small firms are susceptible to storm surge, flooding, and other environmental disasters (Runyan, 2006; Song, Peng, Zhao, &

Hsu, 2016). The overall population of Bay County is 168,852, around 30% of which come from two major coastal cities: Panama City and Panama City Beach (U.S. Census Bureau, 2015).

[Figure]

**Figure 1.** The study area.

[Figure]

**Figure 2.** Historical urban changes (left) and current land uses and zones (right) for the study area.

**2.2 Urban change and zoning**

Figure 2 displays historical urban growth and current land use zoning for the study area. It indicates that urban extent expanded primarily in the southern part of Bay County. As shown in Figure 2, urban developments largely conform to historical trends. Substantial commercial and residential developments occur in two major cities: Panama City Beach and Panama City. In addition, a large piece of land in the north region is zoned for residential uses. This information suggests that local governments and planning agencies have taken measures to encourage inland developments. There exists, however, an apparent discrepancy between zoning and urban growth. Substantial built up areas have appeared in the Towns of Fountain and Youngstown. Nonetheless, such a pattern is not seen on the zoning map where only some areas are designated for the residential use in two towns. This inconsistency suggests that a comprehensive understanding of past land use changes is required to better inform prospective land use zoning.

**2.3 Data description**

**Table 1.** Sources and descriptions of data sets.

| Data type | Year | Spatial resolution | Description and source |
|---|---|---|---|
| Urban extent | 1974 1995 2004 2007 2013 | Vector files | Land use/cover maps from Florida Geographic Data Library |
| Transportation network | 2007 2013 | Vector files | TIGER/Line® Shapefiles and TIGER/Line® Files |
| Slope | - | 30 * 30 m | Converted from National Elevation Datasets of Geospatial |
| Hillshade | - | | Data Gateway from U.S. Natural Resources Conservation Service |
| Exclusion | 1996 2015 | Vector files | Flood Insurance Rate Maps from Florida Geographic Data Library |
| | 2016 | Vector files | Land uses and zones from Bay County GIS online |
| Flooding maps | 2030 2080 | - | The estimates of sea level rise and sea surface temperature published by the IPCC |

Land use and SLR data were prepared for this study. The land use data were comprised of five remotely sensed images on 1974, 1995, 2004, 2007, and 2013; these data were obtained from the Florida Geographic Data Library. These data sets were categorized into nine level-one land cover classes, among which the built-up land was coded as one, according to the Florida land use, cover and forms classification system published by the Florida Department of Transportation (1999). Flooding hazards and zoning information were also collected. Two Flood Insurance Rate Maps (FIRM), developed by the U.S. Federal Emergency Management Agency, were downloaded from the Florida Geographic Data Library. Current zoning map and the comprehensive plan for the study area were obtained from the online GIS websites of Bay County and Washington County. The zoning map specified exactly the degree to which different land uses were allowed for urban developments. According to the Bay County Comprehensive Plan (2009 to 2020), the zoning regulations are shown as follows:

-Developments were not allowed in the zone of conservation (for the preservation purpose), and impervious areas must be no more than 5%;

-In the zone of conservation (for the habitation purpose), impervious surface must be no more than 50%;

-In the zone of conservation (for the recreation purpose), impervious coverage must be no more than 10%; and

5   -In the zones of agriculture (for the timberland purpose) and agriculture, impervious areas must be no more than 10% and 25% respectively.

Finally, the data of projected SLR and sea surface temperatures were collected from the fourth assessment report published by the IPCC (2007). Table 1 displays detailed information regarding all data sets.

**3 Method**

10   ### 3.1 An induction to SLEUTH

**3.1.1 Background**

SLEUTH is a packed C language-based source code that was developed by Dr. Keith C. Clarke at the Department of Geography, University of California Santa Barbara. The source code is freely available through its official website called "Project Gigalopolis" (http://www.ncgia.ucsb.edu/projects/gig/index.html).

15   SLEUTH is comprised of two modules: the Urban Growth Model and the Land Cover Deltatron Model. The Urban Growth Model mainly focuses on the urban/non-urban dynamics. It is more frequently adopted by SLEUTH users than the Land Cover Deltatron Model which investigates changes among different land cover classes. The Urban Growth Model is, therefore, a primary focus of this work. As mentioned, SLEUTH is a CA-based program which only relies on five inputs: urban, transportation, slope, hillshade, and exclusion, and thus it is moderately data driven. The family of CA models has gained more

20   popularity than other modelling techniques. CA models are advantageous over other counterparts due to their spatial explicitness, flexible transitional rules, powerful performance with large data sets (Wagner, 1997), and easy integration with Geographical Information System (Santé et al., 2010). Furthermore, the SLEUTH model was selected for this research because of the following considerations. First, SLEUTH employs excluded layers as probability maps which specify developmental potentials over a study region (where the cell value of zero represents an attracting point for development, and 100 or higher

25   reflects that development is strictly prohibited). Such a functionality makes SLEUTH an excellent platform for scenario-based studies (Leão, Bishop, & Evans, 2004; Jeffrey A. Onsted & Chowdhury, 2014). Second, SLEUTH uses five parameters---dispersion, breed, spread, road gravity, and slope---to establish transition rules which determine whether or not a cell is urbanized. Finally, the calibration process applies a "brute force" approach and is scientifically sound for regional studies (Jeffrey A Onsted & Clarke, 2011).

**3.1.2 SLEUTH workflow**

SLEUTH is a scale-independent CA model that updates the binary state of each cell per growth cycle. A growth cycle is one year and consists of four steps: spontaneous growth, new spreading centres, edge (organic) growth, road-influenced growth (Clarke et al., 1997). These transitional rules are controlled by one or more parameters that were mentioned previously. Each
5 parameter, which has a value range of 1 – 100, is dimensionless and can be compared in terms of their contributions to overall growth. Specifically, the dispersion factor determines the probability by which a cell will be randomly selected for urbanization. The breed factor determines the likelihood by which a newly formed urban cluster will start its own growth cycles. The spread factor controls how likely outward growth will develop near an existing settlement. The road gravity factor demonstrates the influence of road systems upon urban growth by attracting new settlements that are within a certain distance
10 of a road. Finally, the slope factor determines how likely a cell with steeper slope will be urbanized. Table 2 summarizes the relationships between transitional rules and five parameters.

**Table 2.** The relationships among a growth cycle, transitional steps, and controlling factors (Clarke et al., 1997).

| Growth cycle | Transitional steps | Controlling factors | Description |
|---|---|---|---|
| 1-1 | Spontaneous growth | Dispersion | Random urbanization of cells |
| 1-2 | New spreading centers | Breed | Outward growth of new settlements formed in the spontaneous growth stage |
| 1-3 | Edge growth | Spread and slope | Outward growth of current settlements |
| 1-4 | Road-influenced growth | Road gravity, dispersion, breed, and slope | Growth of new settlements within a distance of existing transportation networks |

[Figure]

**Figure 3.** Overall study framework. HUG is historical urban growth; USG is urban sprawl growth; and CUG is compact urban growth.

The main workflow of a SLEUTH application includes input compilations, calibration based on historical urban growth, and predictions. In the Urban Growth Model, at least four maps of different dates which show discernible urban changes are required. Two road networks of different periods and one percentage-slope map are additional date sets. Finally, a hillshade map is used to enhance visualization performance, and water bodies can be embodied in this map. The goal of calibration is to
5    select a combination of parameters that best simulate historical urban changes. This process, however, is enormously time-consuming if all combinations (up to 10 billion) are evaluated. Therefore, a four-stage calibration---coarse, fine, final, and derive----is applied to reduce computational time yet retains acceptable accuracy. Eventually, predictions with 100 Monte Carlo runs are conducted using the best-fit parameters. Full descriptions regarding model inputs, calibration, and predictions can be found in Project Gigalopolis (http://www.ncgia.ucsb.edu/projects/gig/Imp/implement.htm).

10    The workflow for this research was organized into four phases: 1) the preparation of input layers, 2) the calibration in the SLEUTH environment, 3) the predictions of urban growth up to 2080 under the combined scenarios of different developmental patterns and excluded layers, and 4) the overlaying of urban growth estimates and 500-year flooding maps that were induced by SLR. Figure 3 displays the overall research framework. The workflow was divided into two major tasks: the SLEUTH urban growth model and SLR induced flooding maps.

**3.2 Urban growth model**

**3.2.1 Land use/cover related layers**

All input data were processed in ArcGIS 10.3. Five land use maps in the vector format were converted into raster files using the nearest neighbourhood method for spatial resampling. Figure 4 shows urban changes of the study area from 1974 to 2013. Due to the absence of reliable records of transportation networks, only two most recent road maps were used. Because local
20    roads may have very limited influence upon urban growth, only main arteries were extracted from original line files according to the MAF/TIGER Feature Class Code. These polylines were then converted into raster files using the nearest neighbour resampling method. Slope and hillshade were finally generated from the National Elevation Dataset using spatial analyst tool in ArcGIS.

**3.2.2 The creation of E1 excluded layer**

25    An excluded layer reflects the urbanization probabilities of cells. Its cell values range from 0 (unaffected) to 100 (entirely excluded) (Akın et al., 2014). Many publications have investigated how to use excluded layers to enhance calibration accuracy (Rienow & Goetzke, 2015; Sakieh et al., 2015) and evaluate policy scenarios (A. S. Mahiny & Clarke, 2012). Onsted and Clarke (2012) and Akin et al. (2014) recommended that different excluded layers should be used in calibration and prediction. The excluded layer for calibration is suggested to be at minimal restrictions in order to obtain more precise results for urban
30    growth, according to Akin et al. (2014). Hence, this excluded layer (E0) only covers water bodies where urban developments are unrealistic (Figure 4). For prediction phases, however, three excluded layers were applied to represent flooding-risk

mitigation based the whole region (E1), conservational/agricultural land protection (E2), and flooding-risk mitigation based on the Areas More Likely to Experience Growth (AMLEG) (E3).

[Figure]

**Figure 4.** Urban layers, transportation, topographic and historic excluded layer (E1) for model inputs. For the excluded layer, higher value represents higher resistance to urban growth. The value range from 0 to 100.

E1 attempted to assess how likely urban growth appeared in the Special Flood Hazard Area (SFHA) which may be inundated by 100-year floods. Mandatory flood insurance must be purchased by land owners in these areas. Raising construction costs in high-risk regions may partly inhibit vulnerable urban growth. Therefore, E1 could represent a scenario

which guides urban development towards less flood-prone areas. In order to avoid arbitrarily assigning values to excluded layers, Onsted et al. (2014) developed an approach to use historical zoning maps to calculate these values, which was adopted in the E1 excluded layer. Essentially, their method relies on the growth rates whereby urban development occurs in each zone. To retrieve past growth information, we selected 1996 Flood Insurance Rate Map (FIRM) as a reference layer and calculated the area of SFHA and non-SFHA zones as well as the amount of new urban areas from 1995 to 2013 in these zones.

Next, annual rate of urban growth in each zone was determined by Eq. (1):

$$g_n = 1 - ((1 - (\frac{G_n}{Z_n}))^{(1/T)}) \tag{1}$$

where $G_n$ is the total actual urban growth in zone $n$ (1 is SFHA and 2 is non-SFHA zone) from 1995 to 2013,

$Z_n$ is total area of zone $n$ according to the 1996 FIRM, and

$T$ is the number of years, i.e., 18 years.

Finally, the growth rates were used to generate excluded value in the SFHA zone by Eq. (2):

$$E_{SFHA} = 100(1 - (\frac{g_1}{g_2})) \tag{2}$$

where $g_1$ and $g_2$ denote growth rates in SFHA and non-SFHA zone respectively. Table 3 indicates that the growth rate in low-risk areas was approximate three time in SFHA zone, suggesting that mandatory flooding insurance constrained urban expansion in vulnerable regions.

**Table 3.** The development of excluded value for E1 scenario (based on the whole study area).

| Zones | New growth from 1995 to 2013 (hectares) | Total area (hectares) | Growth rate (‰) | Excluded value |
|---|---|---|---|---|
| Special Flood Hazard Area | 665 | 92,054 | 0.4 | 68 |
| Non-SFHA | 4,901 | 216,095 | 1.3 | 0 |
| Water bodies | - | - | - | 100 |

Finally, E1 layer was created based on the 2015 FIRM and represented a managed growth plan that accounted for moderate protection from flooding risks (Figure 5).

**3.2.3 The creation of E2 excluded layer**

The 2020 Bay County Comprehensive Plan was published in 2009 and represented the most recent managed growth option for the study area. Hence, excluded values in the E2 layer were weighted according to this plan and modified based on the work of Akin et al. (2014). Specifically, a cell value of 100 was assigned to water bodies, 95 to the conservation/preservation zones, 50 to the conservation/recreation and agriculture/timberland areas, 25 to other agricultural areas and conservation/habitation zones, and 0 to all other areas (Figure 6).

[Figure]

**Figure 5.** Special Flood Hazard Area in 2015 (left) and the E1 excluded layer (right) (based on the whole study area). For the excluded layer, higher value represents higher resistance to urban growth. The value ranges from 0 to 100.

[Figure]

**Figure 6.** The E2 excluded layer (based on current zoning plans). For the excluded layer, higher value represents higher resistance to urban growth. The value ranges from 0 to 100.

**3.2.4 The creation of E3 excluded layer**

Most SLEUTH modellers apply the above-mentioned methods to develop excluded layers. However, such methods are deficient since they usually treat the whole study area homogeneously. Conversely, urban growth is more likely to involve heterogeneous changes across the study area. For instance, in coastal regions new residential developments largely extend from existing settlements that may only cover a small portion of the whole study region. Thus, Onsted and Chowdhury (2014) developed a procedure that corrects the growth rates in the AMLEG. The authors concluded that the AMLEG technique produced more accurate results than the other two methods: arbitrary guessing and the calculation based on the whole study area. Therefore, the AMLEG was applied based on the E1 scenario (flooding mitigation). The SLEUTH was run in the prediction mode for 100 Monte Carlo times for the period of 1995 to 2013. All five growth coefficients were set as 100, and the cells with an urbanization probability of 50% or more were considered in the AMELG, as shown in Figure 7.

[Figure]

**Figure 7.** The areas more likely to experience growth (AMLEG) in the coastal region from 1995 to 2013. The cells with 50% or more urbanization probability were considered as AMLEG.

Figure 7 shows the coastal areas of high urbanization potentials. The rudimentary simulation of urban growth from 1995 to 2013 justifies the heterogeneous evolution of urban landscape within the whole study region. Thus, the excluded values were re-calculated using the equations (1) and (2), based on the effects of AMLEG (Table 4).

**Table 4.** The growth rates and excluded values for the E3 layer under the effects of AMLEG.

| Zones | New growth from 1995 to 2013 (hectares) | Total area (hectares) | Growth rate (%) | Excluded value |
|---|---|---|---|---|
| AMLEG | 140 | 513 | - | - |
| SFHA | 22 | 98 | 1.4 | 27 |
| Non-SFHA | 118 | 415 | 1.8 | 0 |

The excluded value (27) of the SFHA zone is considerably less than that (68) in the E1 excluded layer. Such a decrease in the excluded value is also found in the study of Onsted and Chowdhury (2014). In addition, this finding indicates that substantial urban growth occurs in flooding-risk areas if the AMLEG effects were considered. This is intuitively reasonable since existing coastal areas are both low-lying places and developmental attractors. Accordingly, the E3 excluded layer was created based on the 2015 FIRM (Figure 8).

[Figure]

Water ■ 27 ☐ 0   Excluded E 3

**Figure 8.** E3 excluded layer (based on the AMLEG technique). The excluded value ranges from 0 to 100. Higher value represents larger resistance to urban growth.

[revised manuscript text omitted]

These growth patterns were next simulated under three policy scenarios: flooding-risk mitigation (E1, based on the whole region), conservational/agricultural land protection (E2), and modified flooding-risk mitigation (E3, based on the AMLEG).

E1 restrained growth in low-lying areas and served as an adaptation strategy to SLR. Alternatively, E2 reflected how future city expansion may be impacted by land use zoning which represented a strong predictor of urban growth in Florida (Jeffrey A. Onsted & Chowdhury, 2014). E3 is a modified scenario of the flooding-risk mitigation based on the fact that urban growth is heterogeneous. Finally, these urban growth predictions were coupled with SLR induced flooding, whose methodology was briefly introduced in the next section.

**3.3 Flooding maps**

The detailed methodology for generating SLR induced flooding were developed by Hsu (2014). In the hurricane model, the effects of rise in sea level and Sea Surface Temperature (SST) were considered in two stages. First, the increase in SST decreased hurricane central pressure over the sea surface (Knutson & Tuleya, 2004). Changed central pressure and other parameters were next used to calculate projected surge heights using the Surge Response Function (SRF) developed by Irish et al. (2009). Based on this information, a hypothetical hurricane was projected to make landfall at a place where it caused the most damages to coastal areas and resulted in a 500-year flood. Second, the projected surge height for the study region was adjusted by local SLR (Udoh, 2012). Two extreme SLR scenarios were considered, as shown in Table 5. A1F1 corresponded to the highest level of global greenhouse gas emissions (IPCC, 2007). Global data were finally adjusted according to local marine conditions in Panama City.

**Table 5.** SST Changes and SLR in Two Future Years in Panama City, FL.

| Climate Scenario | Rise in SST (℃) | SLR (m) |
| --- | --- | --- |
| Present-day | 0 | 0 |
| A1FI (high) in 2030 | 1.23 | 0.20 |
| A1FI (high) in 2080 | 5.02 | 0.90 |

The surge heights were calculated in numerous SRF stations which were defined along the coastline of the study area. SRF zones associated with each station were delineated, and each zone had a height value. Eventually, flooding areas were identified by comparing surge heights and local elevation data.

**4 Results and discussions**

**4.1 Model calibration**

The multi-stage calibration process generated the following parameters: 71 (dispersion), 92 (breed), 70 (spread), 3 (slope), and 35 (road gravity). High values of the first three parameters suggest that past several decades have witnessed apparent urban sprawl and growth surrounding established settlements. On the contrary, low slope value is intuitively reasonable since the case study region barely has mountainous areas, and therefore slope is not a limiting factor. The impact of major roads is rather limited partly because the road network has remained relatively stable since 1980s.

**4.2 Model prediction**

Similar studies have suggested that a sensitivity analysis should be conducted before predictions in order to identify the most significant parameter (Sekovski et al., 2015). The assessment was conducted by subsequently setting each parameter as 80 and keeping the others as the lowest value of 1 and running predictions up to 2030. The results indicate that the spread coefficient has the greatest impact upon future urban expansion, resulting in a 13.99% increase in urban areas up to 2030.

Urban sprawl and compact development can be characterized by different sets of parameters. Specifically, urban sprawl is referred to as scatteredly formed settlements and developments along major transportation networks. Conversely, compact developments are in close proximity to existing urban areas. Therefore, two scenarios, urban sprawl (USG) and compact development (CUG), were developed according to the following criteria.

1. The dispersion, breed, and road gravity coefficients were increased and decreased by 25 in USG and CUG respectively.
2. The spread coefficient was risen and lowered by 10 in USG and CUG respectively. Since this parameter was much more influential than the others in terms of urban growth, 10 was selected as an adjusting value in two scenarios.
3. As its impact was quite marginal, slope parameter remained unchanged across all scenarios. Table 6 summarizes different sets of parameters for three scenarios.

**Table 6.** Growth parameters for three scenarios.

| Scenarios | Dispersion | Breed | Spread | Slope | Road gravity |
| --- | --- | --- | --- | --- | --- |
| Historical growth | 71 | 92 | 70 | 3 | 35 |
| Urban sprawl | 96 | 100 | 60 | 3 | 60 |
| Compact development | 46 | 67 | 80 | 3 | 10 |

We applied different parameter combinations into the SLEUTH model and generated various maps which showed the probability of each cell being urbanized. These maps could be then converted to urban / nonurban results by a cut-off probability value. A justified approach to identify the reasonable cut-off value is to assess the histogram frequency of probabilities (Dezhkam, Amiri, Darvishsefat, & Sakieh, 2013; Rafiee et al., 2009; Wu et al., 2008). After evaluating forecasting maps in 2080, we found that there was a steep increase of urbanized cells around the probability of 90%. The cut-off value of 85, accordingly, was selected to determine whether a cell was converted into urban.

The results show that, under no land use regulations, city areas increase substantially under all scenarios (Figure 9 a). For instance, urban region expands up to 826 km$^2$ in 2080 under the historical growth. Similar patterns can be seen in alternative growth scenarios as well, as shown in Figure 9 a. Under stricter restrictions, the simulations with the E1 excluded layer generate the smallest increase in urban extent from 2013 to 2080 (Figure 9 b). In addition, the growth curves that gradually level off from 2013 indicate a constantly decreasing growth rate. Under compact growth, which shows the most rise in urban areas among three scenarios, the city region expands by 15% within seven decades (Figure 9 b). This is intuitively reasonable because the flooding map exerts a heavy constraint on undeveloped lands, and therefore new developments largely appear near existing settlements. Nonetheless, predictions with the E3 excluded layer produce approximately the same amount of new growth as

those with the E1 layer (Figure 9 d). Similarly, urban growth under the conservational/agricultural land protection (E2) has a similar pattern as simulations with no regulations. Figure 9 c shows that the growth rate in compact development reaches the peak (19%) at 2044 and then gradually levels off. Yet, land use zoning does have an impact upon the amount of new urban areas. The simulated urban area in 2080 with the historical growth pattern is 709 km$^2$ (Figure 9 c), only 85% of that under no restrictions (Figure 9 a). Overall, the results are consistent with similar findings (Sekovski et al., 2015). First, spreading development from existing coastal areas is the leading force behind land use changes. Second, the changes are also driven by dispersion, breed, and road network but less associated with the slope factor.

[Figure]

**Figure 9.** The Simulations of urban changes to the year 2080 of three urban growth patterns under four excluded layers. a) E0: water bodies, b) E1: flooding-risk mitigation (based on the whole region), c) E2: conservational/agricultural land protection, and d) E3: flooding-risk mitigation (based on the AMLEG).

Figure 10 shows predicted urban growth up to 2080 under different excluded layers. These illustrations further depict urban expansion trends indicated by the growth curves in Figure 9. In other words, historical growth and urban sprawl shares similar development patterns where the majority of increased urban cells appear under less strict land use regulations. Moreover, a considerable portion of urban development would be steered towards flooding zones if no land use policy is implemented.

**4.3 The exposure of urban growth to flooding risk**

Figure 11 shows 500-year flooding maps that would be exacerbated by SLR in 2030 and 2080. A vast region along the West, North, and East Bay would be flooded, and the areas immediate to the West and North Bay would be even more susceptible in 2080. As shown in Table 7, the total inundated area in 2080 would be more than 10 times in 2030. Additionally, affected areas with the flooding depth over 3 m would increase exponentially from 2030 to 2080.

[Figure]

**Figure 10.** The comparison of modelled urban extent in 2080 with different excluded layers. E0: water bodies, E1: flooding-risk mitigation (based on the whole region), E2: conservational/agricultural land protection, and E3: flooding-risk mitigation (based on the AMLEG).

[Figure]

**Figure 11.** 500-year flood-risk zones of 0.2-m sea level rise (SLR) (a) and 0.9-m SLR (b).

**Table 7.** Flooded areas under 0.2-m and 0.9-m sea level rise scenarios.

| Flooding depth (m) | Area (km²) | |
| --- | --- | --- |
| | 0.2-m sea level rise (2030) | 0.9-m sea level rise (2080) |
| 0 – 0.5 | 4.6 | 58.8 |
| 0.5 – 1 | 4.3 | 52.5 |
| 1 – 2 | 7.0 | 100.3 |
| 2 – 3 | 7.0 | 77.7 |
| Greater than 3 | 8.2 | 176.1 |
| Total | 31.1 | 465.4 |

Future urban growth simulations were overlaid with flooding maps to show how different development patterns guided by distinct policies are vulnerable to SLR induced flooding (Tables 8 – 9). The results show that, if urban growth progresses compactly, total inundated area in 2030 would be the largest among three growth scenarios (Table 8). This finding is echoed by a previous study (Sekovski et al., 2015). In other words, compact development normally appears surrounding existing urban areas, the majority of which are low-lying and prone to flooding. With respect to land use polices, urban growth under regulations leads to less flood-prone development. Specifically, if no regulations are implemented, on average the flooded area in 2080 would be more than 25 times the area under flooding-risk mitigation (Table 9). Both growth patterns and land use policies, accordingly, substantially affect the susceptibility of coastal cities to flooding hazards.

Figure 12 shows how three urban growth scenarios are exposed differentially to SLR induced flooding at a larger geographical scale. First, urban growth is extremely limited if the excluded layer represents the flooding-risk mitigation strategy based on the whole region. Conversely, if water bodies are used as an excluded layer, urban areas expand considerably in coastlines and hinterlands. Noticeably, urban expansion with the E3 excluded layer also generates a vulnerable landscape to flooding. This reflects the heterogeneous developments over the study region. Coastal areas would probably continue to be

urbanized even if they are jeopardized by flooding and storm surge. This growth pattern may be partly because the high value of properties along shorelines diminishes SLR impacts. Protective structures along coastlines and flooding insurance programs even attract new developments in flood-prone areas. Second, a vast majority of urbanized areas that would be within flooding polygons are situated in the proximity of the West and North Bay and the shoreline areas of Panama City.

**Table 8.** The results of overlaying urban growth predictions with the 500-year flooding map in 2030.

| Excluded layers | Flooded areas | Urban growth scenarios | | |
|---|---|---|---|---|
| | | HUG | USG | CUG |
| E0: no regulations | Total flooded area (m$^2$) | 59,112,900 | 57,466,800 | 59,185,800 |
| | New urban areas that would be flooded (m$^2$) | 7,254,000 | 5,607,900 | 7,326,900 |
| E1: flooding-risk mitigation (based on the whole region) | Total flooded area (m$^2$) | 53,223,300 | 52,974,900 | 54,468,900 |
| | New urban areas that would be flooded (m$^2$) | 1,364,400 | 1,116,000 | 2,610,000 |
| E2: conservational/agricultural land protection | Total flooded area (m$^2$) | 58,479,300 | 56,867,400 | 58,603,500 |
| | New urban areas that would be flooded (m$^2$) | 6,620,400 | 5,008,500 | 6,744,600 |
| E3: flooding-risk mitigation (based on the AMLEG) | Total flooded area (m$^2$) | 57,974,400 | 56,486,700 | 58,048,200 |
| | New urban areas that would be flooded (m$^2$) | 6,115,500 | 4,627,800 | 6,189,300 |

Note: HUG is historical urban growth; USG is urban sprawl growth; and CUG is compact urban growth.

**Table 9.** The results of overlaying urban growth predictions with the 500-year flooding map in 2080.

| Excluded layers | Flooded areas | Urban growth scenarios | | |
|---|---|---|---|---|
| | | HUG | USG | CUG |
| E0: no regulations | Total flooded area (m$^2$) | 200,035,800 | 185,077,800 | 199,561,500 |
| | New urban areas that would be flooded (m$^2$) | 116,387,100 | 101,429,100 | 115,912,800 |
| E1: flooding-risk mitigation (based on the whole region) | Total flooded area (m$^2$) | 87,068,700 | 86,526,900 | 89,570,700 |
| | New urban areas that would be flooded (m$^2$) | 3,420,000 | 2,878,200 | 5,922,000 |
| E2: conservational/agricultural land protection | Total flooded area (m$^2$) | 168,683,400 | 141,254,100 | 164,392,200 |
| | New urban areas that would be flooded (m$^2$) | 85,034,700 | 57,605,400 | 80,743,500 |
| E3: flooding-risk mitigation (based on the AMLEG) | Total flooded area (m$^2$) | 181,416,600 | 155,827,800 | 179,401,500 |
| | New urban areas that would be flooded (m$^2$) | 97,767,900 | 72,179,100 | 95,752,800 |

Note: HUG is historical urban growth; USG is urban sprawl growth; and CUG is compact urban growth.

[Figure]

**Figure 12.** Flooded urban extent in 2080. E0: water bodies, E1: flooding-risk mitigation (based on the whole region), E2: conservational/agricultural land protection, E3: flooding-risk mitigation (based on the AMLEG), HSU: historical growth, USG: urban sprawl, and CUG: compact development

**4.4 Discussion**

**4.4.1 Urban growth and coastal hazards**

The calibration results indicate that main driving force for the study area is spread, followed by dispersion and breed. Such findings are consistent with similar coastal studies (Sakieh et al., 2015; Sekovski et al., 2015). In other words, urban growth is likely to take place around current settlements in a compact fashion. Existing settlements are featured by high accessibility to infrastructure, activity centers, and coastal amenities. Additionally, a multitude of new urban areas cluster around coastlines, which is evident either under historical growth or urban sprawl scenarios. Increased human activities and the competition for limited resources, therefore, intensify environmental pressures at land-sea interfaces. Furthermore, the interface faces unprecedented threats from SLR and other intensified coastal hazards. SLEUTH could provide useful information about future urban growth and thus benefit coastal city managers and land use planners.

However, urban growth models have their own limitations. The most noticeable one is their inability of capturing the whole range of factors affecting urbanization (Herold, Goldstein, & Clarke, 2003). The demand for urban growth comes from population and economic increase. The Bureau of Economic and Business Research at the University of Florida (2016) forecasts that the total population of Bay County will increase by almost 40% by 2030. Such an apparent rise probably demands significant urbanization. Therefore, socioeconomic factors behind urbanization should be considered in applications. Population increase, however, is linked with migration, overall economic conditions, and other factors, which is complicated and hard to predict. Additionally, urbanization in coastal regions is driven by economic activities, the majority of which are related to tourism and real estate. Nevertheless, barely are these factors taken into account in CA models. Third, urban growth predictions may be biased if modellers fail to justify the values in excluded layers. This study determined the excluded values of E2 scenario according to future land use plans and suggestions from other studies (e.g., Akin et al., 2014). Although the lack of historical zoning information forced us to make this assumption, the predictions under the E2 scenario may become problematic.

When urban growth models are coupled with coastal hazards that are associated with SLR, additional concerns arise. In SLEUTH, urban growth predictions fail to incorporate seawall, population relocation, and other adaptation strategies to SLR. This limitation can be seen in the given examples of future flooding risks (Figure 12). Considerable existing urban areas would fall into flooding polygons in 2030 and 2080. Essentially, the model assumes a "do-nothing" option regarding adaptation strategies, which may be improved in future research. Another uncertainty arises during the development of SLR induced flooding. In other words, researchers have not yet reached an agreement as to SLR predictions. Sea level may possibly increase more rapidly than people initially thought. Nicholls and Cazenave (2010) reviewed numerous prediction sources and concluded that global mean sea level would rise between 0.19 m and 1.7 m by 2100. Given these uncertainties, therefore, the simulation results should be interpreted with extreme carefulness and objectivity. In addition, the extrapolation of future urban growth beyond the calibration range can be questionable and generate uncertain results (Goldstein, Candau, & Clarke, 2004). Modellers ought to make a trade-off between urban predictions and the forecasts of climate change related hazards. Climate

change is slow going, but urbanization may be rapid in populated coastal regions. For instance, SLR may become significant only after an adequate time frame that probably exceeds the time period of historical urban data. Such coupled analyses should aim at identifying the general impacts of climate change on prospective urbanization, rather than replicating past patterns of urban development.

**4.4.2 Policy implications**

Compact urban forms are advocated because of their environmental friendliness and energy conservation (Dezhkam et al., 2013; A. Mahiny & Gholamalifard, 2007). Yet, this might not be true in flat coastal areas from the perspective of hazard mitigation. As indicated in Tables 8 and 9, the compact growth model generates more extension of current flood-prone areas than historical growth and urban sprawl. For instance, if the flooding-risk mitigation is implemented, new urban areas that would be flooded in 2030 under the compact growth pattern are over 2,500,000 $m^2$, almost double the area under historical growth scenario. However, such conclusions are made only based on our case study area where the elevation is low and change insignificantly. The effects of compact urban forms on regional exposure to flooding require further investigation in other coastal regions with different topographical features. Therefore, it is recommended that spontaneous urban growth should be regulated to prevent farmland loss, and urban development should be also oriented towards hinterland that already has appreciable urban areas.

Population and economic growth largely drive urban growth. Yet, policies behind such growth should also be investigated since land use plans and economic strategies represent developmental blueprints. Thus, it is beneficial to reflect upon different policies contributing to distinct urban growth patterns: urban sprawl and compact growth. Urban sprawl is characterised by unplanned and scattered developments in suburban areas. Uncoordinated growth in the city edge has been suggested to be associated with multidimensional factors regarding economic incentives, housing development plans, and transportation policies (De Vos & Witlox, 2013; Lopez & Hynes, 2003; Yue, Zhang, & Liu, 2016). Economic incentive packages launched by the central government have contributed to urban sprawl in the developing world. For instance, China took economic reform in 1970s by opening up land markets and commercializing housing units. This economic stimulus gave rise to many sprawling mega-cities such as Beijing, Shanghai, and Chengdu. In Europe and North America, micro-economic theories may explain urban sprawl. Households begin to relocate to the suburbs when the land prices in city centers become prohibitively high. Their relocation decisions are further strengthened by housing and land development policies. Developers promote low-density communities in the city periphery. Local governments help to build large retail centers to accommodate the increased demand. Motorization policies and low fuel costs result in automobile-oriented cities. Even public transit policies aggravate outward city growth by charging long-distance commuters less than riders for short distances (De Vos & Witlox, 2013). As urban sprawl increasingly threatens public health, social equity, and the built environments, people start to develop different urban containment policies. There are primarily two forms of containment policies that were adopted in the US. The State law in Oregon and Washington requires that local land-use plans should clearly define an urban growth boundary. In other states such as Florida and Maryland, governments develop urban service limits, public facilities ordinances, and other policies to promote

compact urban forms (Aytur, Rodriguez, Evenson, & Catellier, 2008). However, the effects of urban containment policies have been hotly debated. For example, not all urban growth boundaries significantly affect housing markets and urbanization paces (Dempsey & Plantinga, 2013). Thus, De Vos and Witlox (2013) suggest the integration of spatial planning policies, mobility policies, and road pricing. Spatial planning can strictly limit new developments outside urban areas. The development

5 of Transit-Oriented Development benefits nurturing high-density and mixed land-use neighbourhoods. In addition, road pricing increases long-distance travel costs, thereby curtailing urban sprawl.

The modelling approach and results offered by this work could aid in the development of an integral land-use enforcement system. Building an integrated land use policy for the urban growth landscape is a recommended option in coastal communities. An integrative policy framework can coordinate the increased demand for urbanization and the goal for hazard mitigation. In

10 addition, land use zoning for existing coastal areas should be incorporated with adaptation strategies to SLR. Rural land use management and regulations should be oriented in order to attract new development inland. While the prohibition of development within flooding zones greatly constrains urban growth and is therefore unrealistic, only relying on land use zoning could lead to considerable inundated urban areas. A compromise of these two alternatives, accordingly, could be developed to ensure adequate urbanization and steer new developments away from low-lying areas. Finally, seawall, planned retreat, and

15 other adaptation strategies should be incorporated in the framework to protect existing urban areas from flooding risks. In the long term, adaptation plans ought to consider other SLR aspects such as groundwater pollution and saltwater intrusion beneath protective structures.

**5 Conclusions**

Environmental and resource pressures are increasingly intensified given ongoing coastal urbanization. In addition, urbanization

20 process amplifies the exposure of coastal communities to flooding hazards. Furthermore, we are uncertain about the degree to which SLR may contribute to the increased intensity of storminess. The possibility of more exacerbated consequences, however, cannot be neglected from a precautionary perspective. Therefore, it is crucial to building an effective coastal management plan to balance land use, competing interests, and hazard mitigation.

[revised manuscript text omitted]

Yue, W., Zhang, L., & Liu, Y. (2016). Measuring sprawl in large Chinese cities along the Yangtze River via combined single and multidimensional metrics. *Habitat International, 57*, 43-52. doi:http://dx.doi.org/10.1016/j.habitatint.2016.06.009

---

## Referee Comment (RC2) · Anonymous Referee #2 · 19 Dec 2016

General Comments The paper entitled "An examination of land use impacts of sea level rise induced flooding" by Song et al. reports on an important analysis that is the assessment of the impact of potential urban developments or landuse of a coastal region on flooding risks associated to sea level rise in future climate. Despite the article focuses on a specific geographic region that means with specific associated risks, lacking in generality, it is overall well-written, well-structured and findings are generally supported by the analysis carried-out. Overall, the article is scientifically sound although I have a number of comments and requests of clarifications as outlined below that in my view need to be addressed by authors to improve the clarity and presentation of some specific aspects. Overall, the article would benefit if a more profound/critical description of choices made for the several steps leading to model outputs were made.

[Figure]

Specific Comments 1)The Abstract is somewhat too qualitative. I suggest to strengthen it to give more emphasis on the methodology used. The SLEUTH model is mentioned without a reference (how this has to be done I guess depends on the specific Editorial formatting procedure) 2)The rationale for the choice of the Bay County has not been addressed. In connection to it the article should give evidence of a larger breath that is how the analysis carried out here could be done in other areas in the world? Despite the research questions clearly states "How would different urban growth patterns increase regional vulnerability to sea level rise induced flooding?" , not enough attention has been paid to why the specific area chosen should be of general interest. The limitations of this study should be clearly stated. 3)The description of the data set (section 2) is rather uncritical. Why these data have been chosen? Are all available data? Would this analysis possible without all these data? 4)Section 3 – I would consider to entitle this section "methodological approach" rather than "method". Please note a typo. An Introduction to the SLEUTH model not "An Induction". The section requires some adjustment. First: please add some references for "dispersion, breed, spread, road gravity, and slope" given that specific definitions of those variables/parameters are application. 5) Overall section 3 is uncritical. The authors report on the method used to set-up the model but fail in explicitly comment on why? For example a function for the annual rate of urban growth (Eq. 1) has been taken that is reasonable but there is no comment on why this should be taken as a general rule or is just a common practice. If so what are the uncertainty associated to given choices? 6)The authors acknowledge the problem of estimating model calibration to reach a good match with data based on metrics. Nevertheless after mentioning the problem they adopt OSM. It would be good to have some comments of the properties/efficacy of such selection. 7)The statistics is used somehow without properly justifying the choices. We read (par. 15, page 9) "Seven Monte Carlo iterations with narrower parameter ranges were employed in the fine stage." Why 7 and not 8, 9, 10. . . what is the impact of this choice? We read "Therefore, a derive calibration with the candidate set were performed with 100 Monte Carlo iterations" Why 100? Can the authors justify and provide more insight on the

choice made? 8)Paragraph 4.1 is interesting but needed to be expanded. 9)As a general remark I suggest wherever possible to point-out that this work is a methodology type of work. Also, the level of approximations, uncertainties associated to each step of the analysis performed are so many that it should be clarified as much as possible that conclusions have to be put in context and somehow used as a general indication of possible risks. 10)Figures overall are of a poor quality. They would also benefit from more substantial captions - at present it is difficult to understand much without a careful reading of the text.

---

## Author Comment (AC2) · 29 Dec 2016

Dear Reviewer,

We are very grateful for your constructive comments that greatly benefit the improvements of our paper. We have paid full attention to all comments and meticulously addressed them. Our responses can be found in the following sections. Please be advised that all line and page pointers (e.g., line 1, page 2, etc.) refer to the revised manuscript that was attached at the end of this document. We have also proofread the manuscript carefully before this submission, which can be shown by another authors' comment posted in the section of interactive discussions. Again, we highly appreciate your time and efforts on reviewing our paper.

1. General comments

1) Despite the article focuses on a specific geographic region that means with specific associated risks, lacking in generality, it is overall well-written, well-structured and findings are generally supported by the analysis carried-out.

Response: we would like to thank the referee for his/her compliments of our paper as well as the concern of the generality of this study. This research selected Bay County in the state of Florida as the study area; however, it does not necessarily indicate that this county of research interests is specifically chosen because of its uniqueness in hazard risks. In other words, it does not imply a lack of generality. We chose Bay County over other coastal areas majorly because it is highly susceptible to coastal flooding and storm surges, and it will be particularly true given future sea level rise. In fact, flooding and storm surges are widespread coastal hazards around the world, and sea level rise has also been observed globally by tide gauges. Many coastal communities in the US and around the world share similar or even higher exposure to such risks (i.e., coastal megacities such as New York and Miami). Thus, we would state that Bay County is somehow representable because it is facing increasing challenges by combined impacts of coastal hydrological hazards and sea level rise that are experienced in many other coastal communities globally. Another principal reason for choosing Bay County is the availability of data needed for modelling.

However, we definitely agree with the referee that we could explain more about the reason why we chose this specific region and how this study could be generalised to other coastal areas. As a result, we have now expanded the discussion of our study area to include why this area was chosen (line 16, page 3) and also provide information on why other coastal communities should be concerned (lines 2 to 5, page 4).

2) Overall, the article is scientifically sound although I have a number of comments and requests of clarifications as outlined below that in my view need to be addressed by authors to improve the clarity and presentation of some specific aspects.

Response: thank you very much for the referee's positive comments about our work.

We have addressed each comment meticulously and illuminated the requests in the following responses and the text as much as possible.

3) Overall, the article would benefit if a more profound/critical description of choices made for the several steps leading to model outputs were made.

Response: thank you for the advice. We made the following overall modifications to justify our choices regarding model inputs and outputs. Specifically, we discussed the rationale of why we chose Bay County as a study area (the reply to specific comment #2). We demonstrated the selection of data inputs, as shown in the response to specific comment #3. We added references and rigorous explanations to support the technical details of calibration, such as the definitions of urban-growth parameters and annual growth rates. These are stated in details in the responses to specific comments # 4, 5, 6, and 7, respectively.

2. Specific comments

1) The Abstract is somewhat too qualitative. I suggest to strengthen it to give more emphasis on the methodology used. The SLEUTH model is mentioned without a reference (how this has to be done I guess depends on the specific Editorial formatting procedure).

Response: we appreciate the referee's suggestions regarding how to ameliorate the Abstract and a comment about the lack of citations. We have substantially improved the Abstract to focus more on the methodological parts: model calibration, prediction, and significant results. We added a reference when the SLEUTH was first introduced (line 10, page 1), and we will work with the Editor to deal with this issue if different procedures should be followed. Specifically, following is a new Abstract (lines 8 to 22, page 1).

"Coastal regions become unprecedentedly vulnerable to coastal hazards that are associated with sea level rise. The purpose of this paper is therefore to simulate prospective

**NHESSD**
[Figure]

urban exposure to changing sea levels. This article first applied the cellular automaton-based SLEUTH model (Project Gigalopolis, 2016) to calibrate historical urban dynamics in Bay County, Florida (US)–a region that is greatly threatened by rising sea levels. This paper estimated five urban-growth parameters by multiple-calibration procedures that used different Monte Carlo iterations to account for modelling uncertainties. It then employed the calibrated model to predict three scenarios of urban growth up to 2080–historical trend, urban sprawl, and compact development. We also assessed land-use impacts of four policies: no regulations; flood mitigation plans based on the whole study region and on those areas that are prone to experience growth; and the protection of conservational lands. This study lastly overlaid projected urban areas in 2030 and 2080 with 500-year flooding maps that were developed under zero, 0.2-m, and 0.9-m sea level rise. The calibration results that a substantial amount of built-up regions extend from established coastal settlements. The predictions suggest that total flooded area of new urbanised regions in 2080 would be more than 25 times that under the flood mitigation policy, if the urbanisation progresses with few policy interventions. The joint model generates new knowledge in the domain between land use modelling and sea level rise. It contributes to coastal spatial planning by helping develop hazard mitigation schemes and can be employed in other international communities that face combined pressure of urban growth and climate change."

2) The rationale for the choice of the Bay County has not been addressed. In connection to it the article should give evidence of a larger breath that is how the analysis carried out here could be done in other areas in the world? Despite the research questions clearly states "How would different urban growth patterns increase regional vulnerability to sea level rise induced flooding?" , not enough attention has been paid to why the specific area chosen should be of general interest. The limitations of this study should be clearly stated.

Response: we are grateful for these comments. We made substantial revisions to the section of the study area and clearly explained why Bay Country was chosen, as

summarised below.

a) Page 3, Lines 16 through Page 4, Line 5: we clarified why Bay County as a particular area can be of general interests and why the results can be generalised to many other coastal regions around the world.

b) Page 4, Lines 6 through 14: we clearly pointed out the region's exposure to sea level rise – one basic rationale for case study selection.

c) Page 4, Lines 15 through 19: we illustrated the data availability issue as another reason for selecting Bay County.

d) Page 23, Lines 8 through 10: we specifically discussed the limitation of this study by adding to this section the following statements. "First, Bay County is a typical land-sea interface confronted with heightened pressure from SLR, and the results are analogous to those in other similar coastal zones. However, we inadequately evaluate the effect of elevation on urban exposure to flooding. Thus, our findings may have limited comparability with hilly areas."

3) The description of the data set (section 2) is rather uncritical. Why these data have been chosen? Are all available data? Would this analysis possible without all these data?

Response: thank you for the comments. We have moved the descriptions of data sets to the sections 3.3 and 3.5, respectively. In the new manuscript, section 3.2 particularly discusses the rationale of data selection for the SLEUTH Urban Growth Model (lines 8 through 11, page 7). Following this is the new section 3.3 that introduce the sources and availability of necessary model inputs. The improved section 3.5 will include the mechanism of the flooding model as well as data requirements and sources (lines 16 through 26, page 15). In this way, the new paper will a better logic flow by integrating the model configurations and rationale and availability of data sets. In response to the last question, the study is fundamentally based on all these data. Specifically, we

addressed the comments in the following aspects.

a) Page 5, Lines 11 through 12: urban, transportation, slope, hillshade, and exclusion are five necessary inputs for the SLEUTH Urban Growth Model.

b) Page 7, Line 9 through 11: we explained why a certain number of maps from different dates are needed for the SLEUTH applications.

c) Page 15, Lines 16 through 26: we illustrated how flooding was influenced by sea level rise in a hurricane model developed in a similar study, what necessary data for modelling are, and where to collect these data.

4) Section 3 – I would consider to entitle this section "methodological approach" rather than "method". Please note a typo. An Introduction to the SLEUTH model not "An Induction". The section requires some adjustment. First: please add some references for "dispersion, breed, spread, road gravity, and slope" given that specific definitions of those variables/parameters are application.

Response: thank you for pointing out the typo and offering suggestions. We have changed the title of section 3 to "methodological approach" (line 2, page 5) and corrected the typo (line 2, page 6). We have also enhanced the logic flow of section 3 by first introducing the overall research framework (line 3, page 5), instead of the background of SLEUTH. We added several references for these five parameters and ensured that we had justifications when using these terms in section 3. Specifically, a reference was added when the parameters were first introduced (line 12, page 6). We also added couples of references in the section 3.2.2 "SLEUTH workflow" to make sure each definition is supported by a reliable source (lines 1, 2, 4, and 4, page 7). Table 1 (line 6, page 7) further gives the relationships between these parameters and four steps for a growth cycle. We made these interpretations based on Clark et al. (1997) who developed the SLEUTH model.

5) Overall section 3 is uncritical. The authors report on the method used to set-up the

model but fail in explicitly comment on why? For example a function for the annual rate of urban growth (Eq. 1) has been taken that is reasonable but there is no comment on why this should be taken as a general rule or is just a common practice. If so what are the uncertainty associated to given choices?

Response: we appreciate the referee for raising concerns about the criticalness of section 3. Although we repeatedly stated the advantages and applicability of the SLEUTH model in section 1 (lines 16-19, page 2), the section 3, and conclusions (lines 10-16, page 17), we agree with the referee that in the methodological part the rationale for the model selection should be first stressed and made very clear. Thus, we made the following improvements and clarifications. First, we enhanced the logic flow of section 3 by first introducing the overall research framework (line 3, page 5), instead of the background of SLEUTH. Following this, we highlighted why the SLEUTH was selected and why it was applicable to our study region (lines 8-15, page 5). Third, the purpose of the annual rate of urban growth is to increase the credibility of weights that correspond to different levels of urbanisation probabilities. This methodology was justified lately (Onsted et al., 2014) and has a great potential to become a general rule in future SLEUTH applications. As suggested by the referee, in the revised manuscript we first explained why we selected this method (lines 1-3, page 10) and then stated its potential values and limitations (lines 4-6, page 10).

6) The authors acknowledge the problem of estimating model calibration to reach a good match with data based on metrics. Nevertheless after mentioning the problem they adopt OSM. It would be good to have some comments of the properties/efficacy of such selection.

Response: we thank the referee for this comment. We justified our selection by adding "The authors evaluated different combinations of the thirteen metrics and found that OSM contributes to more accurate and superior predictions than single-metric approaches. Recent studies have furthermore suggested OSM's robustness (Jantz et al., 2010; Sakieh et al., 2015). Hence, it was applied in this work to narrow parameter

ranges after each stage." (lines 17-20, page 14)

7) The statistics is used somehow without properly justifying the choices. We read (par. 15, page 9) "Seven Monte Carlo iterations with narrower parameter ranges were employed in the fine stage." Why 7 and not 8, 9, 10...what is the impact of this choice? We read "Therefore, a derive calibration with the candidate set were performed with 100 Monte Carlo iterations" Why 100? Can the authors justify and provide more insight on the choice made?

Response: thank you for these comments. The selection of different numbers determines the level of model fit and analytical times. Since SLEUTH applied a "brute force" algorithm, a marginal increase in accuracy is at the expense of exponentially rise in computational time. However, we totally agree with the referee that we should prove our selections appropriately and explicitly in the text. Thus, we added some references to our choices and the following statement.

"While increasing the number of MC iterations can slightly enhance accuracy, the rise in calculation time is extremely pronounced. To balance model fit and efficiency, SLEUTH developers and users experimented in different study areas and developed experiential numbers of MC runs during different steps: 4-5 (coarse); 7-8 (fine); 8-10 (final); and 100 or greater (derive) (Project Gigalopolis, 2016). Hence, this work utilised 4, 7, 9, and 100 MC iterations for each of the four steps respectively. This set is consistent with Sekovski et al. (2015) who examined coastal vulnerability to flooding at a similar geographical scale." (lines 5-10, page 14)

8) Paragraph 4.1 is interesting but needed to be expanded.

Response: we appreciate this comment. We have enriched this subsection by tying the coefficients of calibrated parameters with historical land-use changes. Specifically, we added the following discussions.

"As indicated in Figure 2, the previous urbanisation primarily occurred in the vacant

areas immediate to central Panama City and southwest shorelines. Such an outward expansion of cities is demonstrated by the breed parameter–the most influential factor affecting urban growth. Additionally, two newly urbanised clusters in the north have appeared and been expanding since 1995 (Figure 2). Such a spatial structure is largely captured by the dispersion and breed factors: their values are the second (71) and third (70) highest respectively. By contrast, the low value of the slope parameter is understandable since Bay Country has few mountainous areas, and therefore elevation is not a limiting factor. This finding suggests that the weight of elevation can be further reduced in plain regions, pointing out a direction for customising the data structure of SLEUTH. The road gravity's coefficient is much lower than those of the dispersion, breed, and spread parameters, indicating a limited impact of road systems upon land use allocation. This effect is intuitively reasonable in that transportation networks in the study area have remained stable since the 1980s." (lines 5-15, page 16)

9) As a general remark I suggest wherever possible to point-out that this work is a methodology type of work. Also, the level of approximations, uncertainties associated to each step of the analysis performed are so many that it should be clarified as much as possible that conclusions have to be put in context and somehow used as a general indication of possible risks.

Response: we are grateful for this comment. We have created a new section in the discussion part to talk about modelling and uncertainty issues (line 2, page 23). We also redeveloped and polished the discussion section to stress what the limitations of this work are and what readers should be aware of when employing or interpreting the results. Specifically, we have revised the paper in the following aspects.

a) Page 23, Lines 3 through 7: we stated that this article is majorly a methodological work and that we would talk about the study's limitations related to assumptions and uncertainties.

b) Page 23, Lines 8 through 18: we addressed the referee's concerns regarding the

levels of approximations and the generality of our conclusions.

c) Page 23, Lines 19 through 30: we addressed the referee's comments about the assumptions and their potential risks.

d) Page 23, Line 31 through Page 24, Line 4: in respond to the reviewer's comments, we discussed two aspects of the uncertainties – parameter estimation and the generation of SLR-induced flooding maps.

To address a similar comment from the first referee, we expanded the section 4.4.3 entitled "Policy implications" (line 14, page 24) to deal with the issues of three urban growth scenarios. Since this paper applied three urban forms to represent future land-use dynamics, we made it very straightforward the potential problems of associating urban shapes and their exposure to flooding risks (line 20, page 24). We particularly discussed the relationships between proposed urban forms and policies to improve the practical contributions of this paper (page 24, line 23 through page 25, line 15).

10) Figures overall are of a poor quality. They would also benefit from more substantial captions - at present it is difficult to understand much without a careful reading of the text.

Response: thank you very much for pointing out picture quality issues. We have made significant efforts to enhance the delivery of visual presentations in our manuscript. First, we redesigned almost all figures to increase their readability (please refer to the attached high-resolution figures). Specifically, we increased font size, added important information that was neglected in the original paper, and incorporated subheadings in the figures with subparts. Here, we also would like to thank the first referee for offering perceptive comments about figure problems. Second, we reprocessed all images and optimised visual quality while controlling overall file size. For more information on enhanced figures, please refer to the supplemental materials.

Please also note the supplement to this comment:

http://www.nat-hazards-earth-syst-sci-discuss.net/nhess-2016-157/nhess-2016-157-AC2-supplement.pdf

---

## Author Comment (AC3) · 29 Dec 2016

**An examination of land use impacts of sea level rise induced flooding**

Jie Song[1], Xinyu Fu[1], Yue Gu[1], Yujun Deng[1], Zhong-Ren Peng[1]

[1]Department of Urban and Regional Planning, College of Design, Construction, and Planning, University of Florida, POB 115706, 32611, USA

5  *Correspondence to*: Zhong-Ren Peng (zpeng@ufl.edu)

**Abstract.** Coastal regions become unprecedentedly vulnerable to coastal hazards that are associated with sea level rise. The purpose of this paper is therefore to simulate prospective urban exposure to changing sea levels. This article first applied the cellular automaton-based SLEUTH model (Project Gigalopolis, 2016) to calibrate historical urban dynamics in Bay County, Florida (US)–a region that is greatly threatened by rising sea level. This paper estimated five urban-growth parameters by

10  multiple-calibration procedures that used different Monte Carlo iterations to account for modelling uncertainties.The paper then employed the calibrated model to predict three scenarios of urban growth up to 2080–historical trend,  urban sprawl, and compact development. We also assessed land-use impacts of four policies: no regulations; flood mitigation plans based on the whole study

15  region and on those areas that areMore Likely to experience growth; and the protection of conservational lands. This study lastly overlaid projected urban areas in 2030 and 2080 with 500-year flooding maps that were developed under zero, 0.2-m, and 0.9-m sea level rise . The calibration results  that a substantial amount of built-up regions extend from established coastal settlements. The predictions suggest that

20   total flooded area of new urbanised regions in 2080 would be more than 25 times that under the flood mitigation policy, if the urbanisation progresses with few policy interventions. The joint model generates new knowledge in the domain between land use modelling and sea level rise. it also contributes to coastal spatial planning by helping develop hazard mitigation schemes and can be employed in other international communities that face combined pressure of urban growth and

25  climate change.

**1 Introduction**

Coastal areas are the most intensively exploited places where urban expansion largely alters natural landscape. As land-sea interfaces, however, these regions are featured by various conflicts between anthropogenic pressures and natural sustainability. Moreover, such conflicts have become exacerbated in recent years. While coastal zones increasingly attract

30  population and investments, their communities are more aware of the intensified frequency of natural incidents and possible associations with climate change. It is evident that climate change partly  contribute to intensified hurricanes and floods (Hsu, 2014), rising sea level (IPCC, 2013), and other coastal hazards. More, Sea Level Rise (SLR) may worsen coastal flooding, land submergence, and saltwater intrusion  (Nicholls and Cazenave, 2010). According to the fifth assessment report published by the Intergovernmental Panel on Climate Change (IPCC) in 2013,  approximate

35  70% coastlines will experience rising sea levels by 2100. Nonetheless,  developers, business owners, and other stakeholders will —still continue to compete for limited coastal resources, and the

competition has been even more intense. Developers expedite new real estate projects; local governments offer appealing incentives to attract new infrastructure investments; and  companies extensively extract oil and natural gas in offshore regions (Felsenstein and Lichter, 2014). Since coastal zones are both battlegrounds of conflicting interests and vulnerable low-lying places ,

5 coordinating land uses, hazard mitigation, and different interests has become everlastingly important. Such coordination, therefore, calls for effective tools to inform  coastal management plans. Moreover, spatial planning plays a pivotal role in  these schemes , which can be guided by Land Use/Land Cover Changes (LULCC).

Various techniques can help detect LULCC patterns. The class of Cellular-Automaton (CA) models

10 receive wide attention around the world due to their simplicity and effectiveness in capturing complex urban dynamics (Akın et al., 2014). The operationalization of CA in modelling urban phenomena was first introduced by Clarke et al. (1997) who designed the prototype of SLEUTH–a CA-based urban simulation program. In their model, each cell had a state which updated at consecutive  time points according to predefined  transitional rules. These rules integrated the current condition of a cell,  its neighbours, and  environmental constraints. Numerous

15 software packages have developed since the introduction of CA. Santé et al. (2010) evaluated thirty-three CA models and concluded that the  SLEUTH gained more popularity than the other alternatives. The LEUTH has been adopted by urban researchers all over the world since 2000 (Project Gigalopolis, 2016). Its popularity is partly due to free availability, user-friendliness,  well-developed

20 manuals, and a support forum. An elegant feature of the SLUETH is the application of excluded layers. These layers denote different scenarios in a calibration procedure so as to exemplify how policies influence the expansion of built-up regions (Akın et al., 2014). Recently, SLEUTH was improved by the incorporation of external information incalibration. Rienow and Goetzke (2015) used the support vector machine to enhance  SLEUTH's predictive power by developing probability-based excluded layers. Likewise,

25 Sakieh et al. (2015) applied a multi-criterion evaluation method to generate suitability-based excluded layers.

scenario-based SLEUTH and other CA  models can also forecast future land use, based on historical urban expansion information. By applying a constrained CA model, Hansen (2010) simulated future land conditions under different emission scenarios developed by the IPCC (2013). The author

30 concluded that significant areas in Aalborg would  be increasingly exposed to future flooding hazards.  Hansen (2010) further suggested that more aggressive strategies, such as population relocation and managed retreat, may be evaluated in future simulations. Similarly, Sekovski et al. (2015) assessed the impacts of coastal flooding upon urban growth using SLEUTH model. Inouye et al. (2015) applied a comparative  approach to validate the importance of zoning in land use simulations.

35 By applying a CA package (Dynamic EGO), they stated that the developments in ecological-economic zones heightened the vulnerability to  land sliding,  SLR, and other coastal hazards. Their results greatly benefit the formulation of  relocation, planned retreat, and other adaptation strategies. More specifically researchers also highlighted the importance of zoning  in SLEUTH applications. Akin et al. (2014) used future  zoning maps as excluded layers and

evaluated the accuracy of hindcasting-based calibration. Onsted and Chowdhury (2014) employed a historical zoning map in their SLEUTH model and suggested that land use planningzoning strongly influenced urban growth in Florida.

While the majority of SLEUTH applications focused on different urban forms (e.g., urban sprawl)urban sprawl, other urban forms and, their exposure to flooding received less attention. To the best of our knowledge, only two studies (Garcia and Loáiciga, 2014; Sekovski et al., 2015) attempted to couple land use predictions with marine flooding maps using SLEUTHSLEUTH. The combination of SLEUTH simulations and flooding hazard maps, however, should be prioritised in coastal land use modelling to enhanceinform spatial management (Onsted and Chowdhury, 2014). Therefore, this study aims to evaluate the extent to which SLR-induced flooding will threaten different urban growth patterns using SLEUTH urban growth model may be exposed to SLR-induced flooding. Specifically, two research questions of this study are:

1. How doesmay different urban growth patterns affectincrease coastalregional vulnerability to SLR-induced flooding?
2. Will Would land use zoning and flooding mitigation plans help to steer prospective developments away from low-lyingflood-prone regions?

This paper is organised as follows. Section 2 describes ourthe study area and why this region was selected as a case study and data collection. Following this, section 3 illustrates anthe overall modelling framework, describes data inputs, and outlines major steps for the calibration and prediction of urban growth in the study areacalibration, prediction as well as , and the the development of SLR-induced flooding mapsfloodplain generation. In section 4 we present the calibration coefficients and discuss forecasting outcomes that were overlaid with flooding maps. Finally, section 5 offers a brief conclusion and provides the outlook for future research.

**2 Study area and data description**

**2.1 Background**

This work selected Bay County in Northwest Florida (, USA) , as a primary case study region. Bay County and its adjacent areas (Washington and Walton Counties) were used to provetest our hypotheses based on two considerations: high exposure to coastal hazards and SLR that are experienced in the majority of worldwide coastal zones globally; and data availability for modelling. Bay County has a long shoreline along the Gulf Coast (Figure 1). It is representative in the context of climate change, since it faces unprecedented and accelerated threats from storm surges, hurricanes, and projected sea level variations. Similar exposurevulnerabilities hasve been demonstrated in the European coastline (Vousdoukas et al., 2016), West U.S. Coast (Ludy and Kondolf, 2012), South Vietnam (Apel et al., 2016), and many other coastal megacities around the world.

**Figure 1 about here**

Specifically, Bay County's exposure to marine disastersSLR is pronounced. Historically, Tthise coastal countyzone has been hit by seventeen hurricanes since 1877 (Hurricanecity, 2015). Among these incidences, Hurricane Eloise in 1975 led to enormous $23.1 million damages in structures, seawalls, and patios equivalent up to $23.1 million (Shows, 1978). Moreover, land development patterns render this region extremely susceptible to storm surges and hurricanes and SLR. A considerable amount ofConsiderable residential and commercial buildings structures encroached upon seafront areas in Panama City due to the absence of land use regulations in the past; as a result, urban growth largely occurs in flood-risk zones (Bay County Online, 2016). Primary industrial sectors in Bay Country, as in other coastal communities, are largely susceptible to coastal hazards as well. Bay County

relies on tourism related industries which are deemed vulnerable to  rising sea levels (Ebert et al., 2016).  Bay County has 10,222 firms, of which approximate 90% are small businesses. Unfortunately, Small companies have insufficient resources to cope with storm surge, flooding, and other environmental disasters (Runyan, 2006; Song et al, 2016).

The second rationale for choosing Bay Country is the data variability for modelling. This work used an integrative framework, requiring high-quality data sets regarding land cover and hydrological factors. Land use data can be obtained from different sources . However,  generating SLR-induced flooding maps requires sufficient observations at local and global levels, hurricane records, and many other  marine and meteorological variables—these are available for the study area.  Based on these considerations, we chose Bay County to explore the research questions.

**2.2 Urban change and zoning**

Figure 2 indicates that urban extent expanded primarily in the southern part of Bay County and largely conform to historical trends. A multitude of  residential developments occur in Panama City and shoreline region. In Additionally, many areas in the north is zoned for residential uses. This information suggests that local governments and planning agencies have taken measures to encourage spatially diverse developments. There exists, however, an apparent discrepancy between zoning and urban growth: a substantial amount of urban areas have appeared in  Fountain and Youngstown. Nonetheless, the zoning unclearly reflects this pattern, and  only some areas are designated for the residential land use in the two towns. This inconsistency implies that planners need a comprehensive understanding of past land use changes  to better design zoning schemes.

**Figure 2 about here**
* * *

4

forms classification system published by the Florida Department of Transportation (1999). Flooding hazards and zoning information were also collected. Two Flood Insurance Rate Maps (FIRM), developed by the U.S. Federal Emergency Management Agency, were downloaded from the Florida Geographic Data Library. Current zoning map and the comprehensive plan for the study area were obtained from the online GIS websites of Bay County and Washington County. The zoning map specified exactly the degree to which different land uses were allowed for urban developments. According to the Bay County Comprehensive Plan (2009 to 2020), the zoning regulations are shown as follows:

Developments were not allowed in the zone of conservation (for the preservation purpose), and impervious areas must be no more than 5%;

In the area of conservation (for the habitation purpose), impervious surface must be no more than 50%;

In the zone of conservation (for the recreation purpose), impervious coverage must be no more than 10%; and

In the zones of agriculture (for the timberland purpose) and agriculture, impervious areas must be no more than 10% and 25% respectively.

Finally, the data of projected SLR and sea surface temperatures were collected from the fourth assessment report published by the IPCC (2007). Table 1 displays detailed information regarding all data sets.

**3 Methodological approach**

**FIGURE 3 ABOUT HERE:** Overall study framework. HUG is historical urban growth; USG is urban sprawl growth; and CUG is compact urban growth.

Figure 3 displays the overall research framework. Specifically, the technical The roadmap workflow for this research was organized into four phases: 1) data collection and pre-processing the preparation of input layers, 2) the calibration in the SLEUTH environment, 3) the simulations predictions of urban growth up to 2080 under the combined scenarios of different urban-growth developmental patterns and various excluded layers, and 4) the comparison overlaying of urban growth future urban areas estimates and 500-year flooding maps that were induced by SLR. Figure 3 displays the overall research framework. The framework outline has two major tasks: the SLEUTH urban growth model and SLR-induced flooding maps, as will be discussed in the following sections.

**Figure 3 about here**

**3.1 Rationale for model selection**

We applied SLEUTH as a general n overall modelling architecture based on the following reasons. SLEUTH is a CA-based program which only relies on five inputs: urban, transportation, slope, hillshade, and exclusion, and thus it is moderately data driven. First, The family of CA models has gained more popularity than other modelling techniques. CA models are advantageous over other counterparts due to their spatial explicitness, flexible transitional rules, compatibility powerful performance with large data sets (Wagner, 1997), and easy integration with ArcGIS® Geographical Information System (Santé et al., 2010). Second,

SLEUTH only relies on five inputs: urban, transportation, slope, hillshade, and exclusion. ThirdFirst, SLEUTH employs excluded layers  (where athe cell value of zero represents an attracting point for development, and 100 or higher reflects that urbanizationdevelopment is strictly prohibited). Such a functionality makes SLEUTH an excellent platform for scenario-based studies (Leão et al., 2004).  Finally, itsthe calibration process applies a "brute force" approach and is scientifically sound for regional studies  Onsted and& Clarke, 2011).

**3.2 An introduction to SLEUTH**

**3.2.1 Background**

SLEUTH is a packed C language-based source code that was developed by Dr Keith C. Clarke at the Department of Geography, University of California Santa Barbara. The source code is freely available through its official website called "Project Gigalopolis" (http://www.ncgia.ucsb.edu/projects/gig/index.html).

SLEUTH has two modules: the  Urban Growth Model and the  Land Cover Deltatron Model. The Urban Growth Model mainly focuses on  urban/non-urban dynamics and thus a primary focus of this work.

**3.1.2 SLEUTH workflow**

SLEUTH is a scale-independent CA model that updates the binary state of each cell per growth cycle. A growth cycle is one year and determined by four rules : spontaneous growth, new spreading centres, edge (organic) growth, road-influenced growth (Clarke et al., 1997). These transitional rules are controlled by one or more of the five parameters: dispersion, breed, spread, road gravity, and slope (Project Gigalopolis, 2016). Each parameter, with  a value range of 1 to 100, is dimensionless and can be compared regarding their contributions to overall growth. Specifically, the dispersion factor determines the probability by which a cell will be randomly selected for urbanisation (Silva and& Clarke, 2002). The breed factor determines the likelihood by which a new urban cluster will start its growth cycles (Berberoğlu et al., 2016). The spread factor controls how likely outward growth will develop near an existing settlement. The road gravity factor demonstrates the influence of road systems upon land useurban growth by attracting new developmentssettlements that are within a certain distance of a road (Silva and& Clarke, 2002). Finally, the slope factor calculatesdetermines how likely a cell with a steeper slope will be urbanised (Rafiee et al., 2009). Table 12 summarises the relationships between transitional rules and five parameters.

**Table 1 about here**

The main workflow of a SLEUTH application includes input compilations, a calibration process based on actual urban growth, and predictions. The Urban Growth Model requires at least four maps of different dates which show obvious urban changes. Two road networks of different periods and one percentage-slope map are additional date sets. An optional hillshade map is used to improve visualisation performance. The goal of the calibration is to select a combination of the parameters that best replicate historical urban changes. This process, however, is enormously time-consuming if modellers assess all combinations–up to 10

billion. Therefore, SLUETH applies a four-stage calibration process (coarse, fine, final, and derive) to reduce computational time. FinallyEventually, the predictions with 100 Monte Carlo (MC) runs are conducted using the best-fit parameters.

FIGURE 3 ABOUT HERE: Overall study framework. HUG is historical urban growth; USG is urban sprawl growth; and CUG is compact urban growth.

The workflow for this research was organized into four phases: 1) the preparation of input layers, 2) the calibration in the SLEUTH environment, 3) the predictions of urban growth up to 2080 under the combined scenarios of different developmental patterns and excluded layers, and 4) the overlaying of urban growth estimates and 500-year flooding maps that were induced by SLR. Figure 3 displays the overall research framework. The outline has two major tasks: the SLEUTH urban growth model and SLR-induced flooding maps.**3.3 Data description** for the Urban Growth Model

Table 2 displays detailed information regarding all data sets required by the Urban Growth Model. The land use data were obtained from the Florida Geographic Data Library (FGDL) and included five remotely sensed images in 1974, 1995, 2004, 2007, and 2013. These data sets were categorised into nine level-one land cover classes, among which the built-up land was coded as one, according to the Florida land use, cover and forms classification system published by the Florida Department of Transportation (1999). Flooding hazard and zoning information were also collected to create excluded layers. Two Flood Insurance Rate Maps (FIRM), developed by the U.S. Federal Emergency Management Agency, were downloaded from the FGDL. Current zoning map and the comprehensive plan for the study area were obtained from the online GIS websites of Bay and Washington Counties. The zoning map specified the degree to which different land uses were allowed for urban developments. According to the Bay County Comprehensive Plan (2009 to 2020), the zoning regulations are shown as follows:

-Developments were not allowed in the zone of conservation (for the preservation purpose), and impervious areas must be no more than 5%;

-In the area of conservation (for the habitation purpose), impervious surface must be no more than 50%;

-In the zone of conservation (for the recreation purpose), impervious coverage must be no more than 10%; and

-In the zones of agriculture (for the timberland purpose) and agriculture, impervious areas must be no more than 10% and 25% respectively.

**Table 2 about here**

**3.42 Urban Growth Model**

**3.42.1 Land use/cover related layers**

All input data were processed in ArcGIS® 10.3. Five land use maps in the vector format were converted into raster files using the nearest neighbourhood method. Figure 4 shows urban changes in Bay County from 1974 to 2013. Only two most recent road-network maps were used due to data limitations. Because local roads may have a very limited influence upon urban growth, only main arteries were extracted from original line files according to the MAF/TIGER Feature Class Code. These polylines were then converted into raster files using the nearest neighbour resampling method. The sSlope and hillshade maps, collected from the National Elevation Dataset, were finally generated from the National Elevation Dataset using the spatial analyst tool in ArcGIS.

**3.42.2  The creation of E1 excluded layer**

An excluded layer reflects the urbanisation probabilities of cells. Its cell values range from 0 (unaffected) to 100 (entirely excluded) (Akın et al., 2014). Many publications have investigated how to use excluded layers to enhance calibration accuracy (Rienow & Goetzke, 2015; Sakieh et al., 2015) and evaluate policy scenarios (A. S. Mahiny & Clarke, 2012). Onsted and Clarke (2012)  and Akin et al. (2014) recommended that the calibration and prediction stages utilise different excluded layers. The excluded layer for the calibration is suggested to be at minimal restrictions to obtain more precise results offor urban growth, according to Akin et al. (2014). Hence, this excluded layer (E0) only covers water bodies where urban developments are unrealistic (Figure 4). For the prediction phases, however, three excluded layers were applied to represent flood ing-risk mitigation based the whole region (E1), conservational/agricultural land protection (E2), and flooding-risk mitigation based on the Areas More Likely to Experience Growth (AMLEG) (E3).

**Figure 4 about here.**

**FIGURE 4 ABOUT HERE:** Urban layers, transportation, topographic and historical excluded layer (E1) for model inputs. For the excluded layer, the higher value represents higher resistance to urban growth. The value ranges from 0 to 100.

The E1 attempted to assess how likely urban growth appeared in the Special Flood Hazard Area (SFHA) which may be inundated by 100-year floods. Landowners must purchase mandatory flood insurance in these areas. Increasing construction costs in high-risk regions may partly inhibit vulnerable urban growth. Therefore, the E1 could represent a scenario which guides urban development towards fewer flood-prone areasprohibits land developments in floodplains.

Weights in the E1 layer were determined according to the approach which Onsted et al. (2014) used to reduce the errorsbias of randomly assigned values. To avoid arbitrarily assigning values to excluded layers, Onsted et al. (2014)the authors elegantly used historical zoning maps to calculate the weights. Their method bases onrelies on annual the growth rates whereby new urban developments appear occurs in differenteach zones. This method has a great potential to become a general rule in future SLEUTH applications. However, it assumes the annual rate remains unchanged over timea linear relationship between urban growth rates and time series. Therefore, concerns may arise if a study area experiences non-linear growing growth rates over a period, which should be addressed in future studies. It is therefore suggested that modellers modify the Methodological assumptions when applying the method to these study regions.

To retrieve past growth information, we selected the1996 FIRMlood Insurance Rate Map (FIRM) as a reference layer and calculated the area of SFHA and non-SFHA zones as well as the amount of new urban areas from 1995 to 2013 in these zones respectively.

Next, the annual rate of urban growth in each zone was determined by Eq. (1):

$$g_n = 1 - ((1 - (\frac{G_n}{Z_n}))^{(1/T)}) \tag{1}$$

where $G_n$ is the total actual urban growth in zone $n$ (1 is the SFHA and 2 is the non-SFHA zone) from 1995 to 2013,

$Z_n$ is total area of zone $n$ according to the 1996 FIRM, and

$T$ is the number of years, i.e., 18 years.

Finally, Tthe growth rates were used to generate  excluded value in the SFHA zone by Eq. (2):

$$E_{SFHA} = 100(1 - (\frac{g_1}{g_2}))$$

(2)

where $g_1$ and $g_2$ denote  growth rates in  SFHA and non-SFHA zones respectively. Table 3 indicates that the growth rate in low-risk areas was approxima three times  in  SFHA zone, suggesting that mandatory floo insurance

5   constrained urban expansion in flood-prone regions. Finally, the E1 layer was created based on the 2015 FIRM and represented a growth management that aims at mitigating flood risks (Figure 5).

Table 3 about here

10

**3.4.3 The creation of E2 excluded layer**

15   The 2020 Bay County Comprehensive Plan was published in 2009 and represented the most recent managed growth option for the study area. Hence, excluded values in the E2 layer were weighted according to this plan and modified based on the work of Akin et al. (2014). Specifically, a cell value of 100 was assigned to water bodies, 95 to the conservation/preservation zones, 50 to the conservation/recreation and agriculture/timberland areas, 25 to the other agricultural areas and conservation/habitation zones, and 0 to all other areas (Figure 6).

**Figures 5 and 6 about here**

25

30   **3.4.4 The creation of E3 excluded layer**

Most SLEUTH modellers apply the above-mentioned methods to develop excluded layers. However, such approaches may be deficient since these  treat the whole study area homogeneously. Realistically, urban growth involve heterogeneous changes across the study area. For instance,  new residential developments largely extend from existing settlements that may only cover a small area
35   within a city boundary. Thus, Onsted and Chowdhury (2014) developed a procedure that corrects the growth rates in the AMLEG. The authors concluded that the AMLEG technique produced more accurate results than the other  methods: arbitrary guessing and the calculation based on the whole study area. Therefore, the AMLEG approach was applied  based on the E1 scenario

(flood ing mitigation). Specifically, The SLEUTHSLEUTH was first run in athe prediction mode with for 100 MCMonte Carlo times for the period of 1995 to 2013. All five growth coefficients were set as 100, and the cells with an urbanisation probability of 50% or more were considered in the AMELG (, as shown in Figure 7). Second, excluded values in the E3 excluded layer were re-calculated using the equations (1) and (2), based on the AMLEG effects (Table 4). Finally, Accordingly, the E3  excluded layer was created based on the 2015 FIRM (Figure 8).

**Figure 7 about here**

**Table 4 about here**

Figure 7 ABOUT HERE: The areas more likely to experience growth (AMLEG) in the coastal region from 1995 to 2013. The cells with 50% or more urbanisation probability were considered as AMELG.

Figure 7 shows the coastal areas of high urbanisation potentials. The rudimentary simulation of urban growth from 1995 to 2013 justifies the heterogeneous evolution of urban landscape within the whole study region. Thus, the excluded values were re-calculated using the equations (1) and (2), based on the effects of AMLEG (Table 4).

TABLE 4 ABOUT HERE

Figure 7 shows the coastal areas with high-urbanisation potentials. The rudimentary simulation of urban growth from 1995 to 2013 justifies the heterogeneous evolution of urban landscape within the whole study region. The excluded value (27) of the SFHA zone is considerably less than that (68) in the E1 excluded layer. Such a decrease was also discovered by Onsted and Chowdhury (2014) also discovered such a decrease in the excluded value. Also, this finding indicates that substantial urban growth occurs in flooding-prone areas if we considered the AMLEG effects. This phenomenon is intuitively reasonable since existing coastal regions are both low-lying places and developmental attractors.

**Figure 8 about here**

Accordingly, the E3 excluded layer was created based on the 2015 FIRM (Figure 8).

FIGURE 8 ABOUT HERE. E3 excluded layer (based on the AMLEG technique). The excluded value ranges from 0 to 100. A higher value represents larger resistance to urban growth.

All pre-processed these raster files, including land use maps and four excluded layers, were eventually resampled at a spatial resolution of 30 m * 30 m, which is adequately high since the resolutions of most SLEUTH applications are in the range of 10-100m (Akın et al., 2014). These raster data files with 1972 rows * 2383 columns were then exported as grayscale GIF images and imported into the the SLEUTHSLEUTH program.

**3.2.5 Model calibration**

As mentioned, urban-growth parameters were calibrated based on the "brute force" technique where analysts follow four calibration stages—coarse, fine, final, and derive. Increasingly higher image resolutions were typically used from the coarse to final calibrations for computational efficiency (Akın et al., 2014; Chakraborty et al., 2015; Rafiee et al., 2009). However, different resolutions may be problematic and lead to a biased estimation of growth patterns(Dietzel and Clarke, 2007). Thus, the consistent resolution of 30 * 30 m was employed during the entire calibration process.

In each calibration phase, several MC iterations were simulated to account for the uncertainty associated with parameter estimations (Project Gigalopolis, 2016). A general strategy of identifying the "best-fit" parameters is to shrink the range of parameters during each phase. While increasing the number of MC iterations can slightly enhance accuracy, the rise in calculation time is extremely pronounced. To balance model fit and efficiency, SLEUTH developers and users experimented in different study areas and developed experiential numbers of MC runs during different steps: 4-5 (coarse); 7-8 (fine); 8-10 (final); and 100 or greater (derive) (Project Gigalopolis, 2016). Hence, this work utilised 4, 7, 9, and 100 MC iterations for each of the four steps respectively. This set is consistent with Sekovski et al. (2015) who examined coastal vulnerability to flooding at a similar geographical scale.

Four MC runs were conducted in the coarse calibration, and the widest range of the parameters, 0 to 100, were evaluated with an increment of 25 at a time. The goodness of fit in models was assessed by thirteen metrics, the majority of which were least-square regression scores between simulated urban components (e.g., increased urban pixels and clusters) and real counterparts. SLEUTH scholars, however, largely debated the selection of the optimal metric which could determine model performance. Most disputes centred on whether a combination of several indicators outperform a single metric. Dietzel and Clarke (2007) developed a composite metric, known as Optimal SLEUTH Metric (OSM). The OSM is the product of the compare, population, edges, clusters, slope, X-mean, and Y-mean metrics. The authors evaluated different combinations of the thirteen metrics and found that OSM contribut to more accurate and superior predictions than single-metric approaches. Recent studies have furthermore suggested OSM's robustness (Jantz et al., 2010; Sakieh et al., 2015). Hence, it was applied in this work to narrow parameter ranges after each stage. Specifically, seven MC iterations with narrower parameter ranges were employed in the fine stage. Further refined ranges of the parameters with nine MC iterations were next tested during the final calibration. This whole process took around one-month process time and was conducted using the standard 3.0 SLEUTH model executed in the Cygwin UNIX Windows compiler. This three-stage process generated five candidate parameters; however, this set may be biased due to the self-modification nature of SLEUTH. Therefore, a "derive" calibration with the candidate set was performed with 100 MC iterations.

**3.2.6 Model prediction**

Three approaches have been widely applied in the literature. The first is to adjust one or more of the five parameters (Leao et al., 2004; Rafiee et al., 2009(Leao et al., 2004; Rafiee et al., 2009; Sekovski et al., 2015). The second is to modify the growth-resistance levels in excluded layers (Jantz et al., 2010) or apply distinct excluded layers (Akın et al., 2014). The last one, less frequently used than the first two methods, is to alter self-modification parameters which control overall growth rates (Yang and Lo, 2003). This research used a combination of the first two approaches, and the results were overlaid with SLR-induced flooding maps. The urban growth was predicted up to 2080, and the exposure to flooding under different growth patterns and policies were analysed in 2030 and 2080.

As in widespread use in similar studies, this work first simulated three growth patterns: Historical Growth (HUG), Urban Sprawl (USG), and Compact DevelopmentUrban Growth (CUG). The HUG assumed that prospective urban extent expanded at an existing growth -rates, and . fFive parameters remained unchanged over the forecasting period. The USG resulted in more scattered urban communities which appeared in the suburbs and along transportation corridors. In the model, the dispersion, breed, and road gravity factors controlled sprawling growth (Table 2), so increasing these parameters produced more dispersed developmentsgrowth. On the contrary, the CUG greatly is characterised byrelied on the expansionenlargement of current settlements. Compact development is is apparent in Bay Country the study area and many other populated coastal regions. In SLEUTH, compact development was positively associated with the parameter, spread parameter (Table 2). In addition, decreasing road gravity inhibited new growth along corridors and thus contributed to compact urban forms (Table 2).

These growth patterns were next simulated under three policy scenarios: flood ing risk mitigation (E1, based on the whole region), conservational/agricultural land protection (E2), and modified flooding risk mitigation (E3, based on the AMLEG). E1 restrained growth in low-lying areas and served as an adaptation strategy to SLR. Alternatively, E2 reflected how future city expansion may be impacted by land use zoning thatwhich representsed a strong predictor of urban growth in Florida (Jeffrey A. Onsted and& Chowdhury, 2014). E3 is a modified scenario of the flooding risk mitigation due to heterogeneous based on the fact that urban growth is heterogeneous. Finally, these urban growth predictions were coupled with SLR-induced flooding. , whose methodology was briefly introduced in the next section.

**3.53 SLR-induced fFlooding maps**

The detailed methodology for generating SLR-induced flooding was developed by Hsu (2014). In his hurricane model, the effects of the rise in sea level and sSea sSurface tTemperature (SST) were considered in two stages. First, the increase in SST decreased central hurricane pressure over the sea surface (Knutson and& Tuleya, 2004). Changed central pressure and other parameters were next used to calculate projected surge heights using the sSurge rResponse fFunction (SRF) developed by Irish et al. (2009). Based on this information, a hypothetical hurricane was projected to make landfall at a place where it caused the most damages to coastal areas and resulted in a 500-year flood. Second, the projected surge height for the study region was adjusted by local SLR (Udoh, 2012). Two extreme SLR scenarios were considered, as shown in Table 5. A1F1 corresponded to the highest level of global greenhouse gas emissions (IPCC, 2007). Global data were finally adjusted according to local marine conditions in Panama City. Next, different the surge heights were calculated in numerous SRF stations which were defined along the coastline of Bay Countythe study area. SRF zones associated with each station were delineated, and each zone had a height value. FinallyEventually, flooding areas were identified by comparing surge heights and local elevation data. All the required data sets regarding projected SLR and SST were collected from the fourth assessment report published by the IPCC (2007) (Table 5).

Table 5 about here

TABLE 5 ABOUT HERE

**4 Results and discussions**

**4.1 Model calibration**

The multi-stage calibration process generated the following parameters: 71 (dispersion), 92 (breed), 70 (spread), 3 (slope), and 35 (road gravity). High values of the first three parameters suggest that the past several decades have witnessed apparent urban sprawl and growth surrounding established settlements. As indicated in Figure 2, the previous urbanisation in the past several decades primarily occurred in the vacant areas immediate to central Panama City and southwest shorelines. Such an The considerably outward expansion of cities is demonstrated by the breed parameter—the most influential factor affecting urban growth. Additionally, two newly urbanised clusters in the north have appeared and been expanding since 1995 (Figure 2). Such a spatial structure is largely captured by the dispersion and breed factors: theirwhose values are the second (71) and third (70) highest respectively. By contrastOn the contrary, the low value of the slope parameter value is understandable since Bay Countrythe case study region barely has few mountainous areas, and therefore elevation the slope is not a limiting factor. This finding suggests that the weight of elevation can be further reduced in plain regions, pointing out a direction for customising the data structure of SLEUTH. The road gravity's coefficient is much lower than those of the dispersion, breed, and spread parameters, indicating a limited impact of road systems upon land use allocation. This effect is intuitively reasonable in that transportation networks in the study area have remained stable since the 1980s.

**4.2 Model prediction**

It isSimilar studies have suggested that a sensitivity analysis should be conducted before predictions to identify the most significant parameter (Sekovski et al., 2015). Thise assessment was carried out by subsequently setting each parameter as 80 and keeping the others as the lowest value of one1 and running predictions up to 2030. The results indicate that the "spread" parametercoefficient has the greatest impact on future urban expansion, leading to a 13.99% increase in urban areas up to 2030.

Different sets of parameters can characterise urban sprawl and compact development. Specifically, urban sprawl is referred to as scatteredly formed settlements and developments along major transportation networks. Conversely, compact developments are in proximity to existing urban areas. Therefore, this work applied the following criteria to develop two alternative scenarios of future urban growth scenarios, urban sprawl (USG) and compact development (CUG), were developed according to the following criteria.

1. The dispersion, breed, and road gravity's coefficients coefficients were increased and decreased by 25 in USG and CUG respectively.

2. The "spread's" coefficient was risen and lowered by 10 in USG and CUG respectively. Since this parameter was much more influential in urban growth than the otherss regarding urban growth, ten was selected as an adjusting value in two scenarios.

3. As its impact was quite marginal, the slope parameter remained unchanged across all scenarios. Table 6 summarises different sets of parameters for three scenarios.

**Table 6 about here**

**TABLE 6 ABOUT HERE**

We applied different parameter combinations into  the SLEUTH model and generated various maps which showed the probability of each cell being urbanised. These maps could be then converted to urban/nonurban results by a cut-off probability value. Here, a justified approach to identify the reasonable cut-off value is to assess the histogram frequency of probabilities (Dezhkam et al., 2013; Rafiee et al., 2009; Wu et al., 2008). After evaluating the projected maps in 2080, we found that there was a steep increase of urbanised cells around the probability of 90%. The cut-off value of 85  was selected to determine whether a cell was converted into urban accordingly.

Figure 9 summarises the growth statistics with different policies under three urban development scenarios. It  show that, without land use regulations, city areas grow substantially under all scenarios (Figure 9 a). For instance, urban region expands up to 826 km$^2$ in 2080 under the historical growth. Similar patterns can be seen in alternative growth scenarios as well, as shown in Figure 9 a. Under stricter land use restrictions, the simulations with the E1 excluded layer generate the smallest increase in urban extent from 2013 to 2080 (Figure 9 b). In addition, their growth curves that gradually level off from 2013 indicate a steadily falling growth rate. Under compact growth, which shows the highest  rise in urban areas among three scenarios, the city region expands by 15% within seven decades (Figure 9 b). This is intuitively reasonable because the flooding map exerts a heavy constraint on undeveloped lands, and therefore new developments largely appear near existing cities and towns. Nonetheless, the predictions with the E3 excluded layer produce approximately the same amount of new growth as those with the E1 layer (Figure 9 d). Similarly, urban growth with the E2 layer ( conservational/agricultural land protection) has a similar pattern as the simulations with no regulations. Figure 9 c shows that the growth rate in compact development reaches the peak (19%) at 2044 and then gradually levels off. However, land use zoning does have an impact on the amount of new urban areas. The simulated urban area in 2080 with the historical growth pattern is 709 km$^2$ (Figure 9 c), only 85% of that under no restrictions (Figure 9 a). In sum, spreading development from existing coastal areas is the leading force behind land use changes. Second, the changes are also driven by the dispersion, breed, and road parameters but less associated with the slope factor.

**Figure 9 about here**

Figure 10 shows predicted urban growth up to 2080 under different excluded layers. These illustrations further depict urban expansion trends indicated by the growth curves in Figure 9. In other words, historical growth and urban sprawl share similar developmental patterns where the majority of projected urban cells appear under flexible land use regulations. Moreover, a considerable portion of urban expansion would fall into flooding zones if no land use policy is implemented.

**Figure 10 about here**

**4.3 The exposure of urban growth to flooding risk**

Figure 11 shows 500-year flooding maps that would be exacerbated by SLR in 2030 and 2080. A Large region immediate to the West, North, and East Bays would be flooded, and the regions adjacent  to the West and North Bays would

be even more susceptible in 2080. As shown in Table 7, the total inundated area in 2080 would be more than ten times that in 2030. Additionally, the total amount of lands with a flooding depth over 3 m would rise considerably from 2030 to 2080.

5

Future urban simulations were overlaid with flooding maps to show how different developmental patterns guided by distinct policies are vulnerable to SLR-induced flooding (Tables 8–9). The results unveil that, if urban growth
15  progresses compactly, total inundated area in 2030 would be the largest among three growth scenarios (Table 8).  For land use policies, urban growth under regulations leads to less flood-prone developments. Specifically, if no regulations are implemented, the total inundated area of projected urbanised cells in 2080 is 111.243 km$^2$ on average, more than 25 times the area in the flooding-risk mitigation strategy based on
20  the whole region (Table 9). Therefore, both growth patterns and land use policies have substantial impact on the susceptibility of coastal cities to flooding hazards.

Figure 12 shows how three urban growth scenarios are exposed differentially to SLR-induced flooding at a larger geographical scale.  urban growth is extremely limited if we implement the excluded layer that represents the floodrisk mitigation  based on the whole region. Conversely, if water bodies are used as an excluded layer, urban areas expand
25  considerably in coastlines and hinterlands. Noticeably, urban expansion with the E3 excluded layer also generates a vulnerable landscape to flooding. This phenomenon reflects  heterogeneous developments over the study region. Coastal areas would probably continue to be urbanised even if they are threatened by flooding and storm surge. Such a pattern may be partly because the high value of properties along shorelines diminishes SLR impacts. Protective structures  and flood insurance programs even attract new developments in flood-prone areas.
30

35  **Figure 12 about here**

**4.4 Discussion**

**4.4.1 Modelling and uncertainties**

According to Box and Draper (1987), "Essentially, all models are wrong, but some are useful" (p. 424).  simulations are different levels of the approximations of real-world phenomena. This norm also applies to this work. The primary contribution of this paper is to seek methodological incorporation between urban growth models and SLR-induced flooding, although it attempts to tie land use predictions with coastal planning practice. However, this research suffers from a few limitations associated with the case study region, assumptions,  uncertainties.

Frist, Bay County is a typical land-sea interface confronted with heightened pressure from SLR, and the results are analogous to those in other similar coastal zones. However, we inadequately evaluate the effect of elevation on urban exposure to flooding . Thus, our findings may have limited comparability with hilly areas.

Second, the SLEUTH model in its current form excludes several  critical variables contributing to urban dynamics. It may inadequately capture various factors affecting urbanisation (Herold et al., 2003). The demand for urban growth comes from population and economic increase. The Bureau of Economic and Business Research at the University of Florida (2016) forecasts that the total population of Bay County will increase by almost 40% by 2030. Such a rise  demands additional urbanisation capacities. Therefore, future applications should consider socioeconomic factors behind urbanisation . Population growth, however, is linked with migration, overall economic conditions, and other factors, which is complicated and hard to predict. Additionally, urbanisation in coastal regions is driven by economic activities, the majority of which are related to tourism and real estate. Nevertheless, barely do CA models integrate  these factors .

Third, this paper bases on couples of assumptions to forecast urban landscape evolution. It determines the excluded values of the E2 scenario according to future land use plans and suggestions from other studies (e.g., Akin et al., 2014). Although the lack of  zoning information forced us to make this assumption, the predictions under the E2  may become problematic. Besides, the extrapolation of future urban growth beyond the calibration range can be questionable and generate uncertain results (Goldstein et al., 2004). Modellers ought to make a trade-off between land use predictions and the projections of climate change related hazards. Climate change is slow going, but urbanisation may be rapid in populated coastal regions. For instance, SLR may become significant only after an adequate period that probably exceeds that of historical urban data. Such coupled analyses should aim at identifying the general impacts of climate change on future urbanisation, rather than replicating the past urban patterns . The last assumption relates to adaptation strategies.  SLEUTH  fails to incorporate seawall, population relocation, and other adaptation strategies to SLR. This limitation can be seen in the given examples of future flooding risks (Figure 12). Many existing urban areas would fall into flooding polygons in 2030 and 2080. Essentially, the model assumes a "do-nothing" option regarding adaptation strategies; such a limitation is addressed in a following paper .

Eventually, uncertainties come from two aspects. The "best-fit" parameters are non-deterministic and estimated by distinct numbers of MC iterations in different calibration stages. Thus, when comparing this work's numerical results with those of similar research, readers should be aware of the stochastic nature of the model. Second, addition concerns arise when it comes to SLR projections and their impacts on hurricanes. Researchers have not yet reached an agreement as to SLR estimates; Sea level may increase more rapidly than people initially thought. Nicholls and Cazenave (2010) reviewed numerous  sources and concluded that globally mean sea level would rise between 0.19 m and 1.7 m by 2100. Given these uncertainties, therefore, the simulation results should be interpreted with caution.

**4.4.2 Urban growth and coastal hazards**

The calibration results indicate that main driving force for the study area is spread, followed by dispersion and breed. Such findings are in line with similar coastal studies (Sakieh et al., 2015; Sekovski et al., 2015). In other words, urban growth is likely to take place around current settlements in a compact fashion. Existing settlements are featured by excellent accessibility to infrastructure, activity centres, and coastal amenities. Additionally, a multitude of new urban areas clusters around coastlines, which is evident either in the business-as-usual or urban sprawl scenarios. Increased human activities and the competition for limited resources, therefore, intensify environmental pressures at land-sea interfaces. Furthermore, the interface faces unprecedented threats from SLR and other intensified coastal hazards. SLEUTH could provide useful information about future urban growth and thus benefit coastal city managers and land use planners.

**4.4.3 Policy implications**

Compact urban forms are advocated because of their environmental friendliness and benefits for energy conservation (Dezhkam et al., 2013; Mahiny and Gholamalifard, 2007). However, this might not be true in flat coastal areas from the perspective of hazard mitigation. As indicated in Tables 8 and 9, the compact growth scenario generates more extension of current flood-prone areas than the scenarios of historical growth and urban sprawl. For instance, if the flood mitigation policy is implemented, new built-up regions that would be flooded in 2030 under the compact growth pattern are over 2.5  km$^2$, almost double the area under historical growth scenario. However, such conclusions are made only based on our case study area where slopes change insignificantly. Thus, the effects of compact urban forms on flooding exposure require further investigation in other coastal regions with different topographical features.

Population and economic growth largely drive urban expansion. Nevertheless, policies behind such growth should also be investigated since these policies (e.g., land use plans and economic strategies) represent developmental blueprints. Thus, it is beneficial to reflect upon how policies contribute to distinct urban growth patterns: urban sprawl and compact growth. Urban sprawl is characterised by unplanned and scattered developments in suburban areas. Uncoordinated growth in the city edge has been suggested to relate to  multidimensional factors regarding economic incentives, housing development plans, and transportation policies (De Vos and Witlox, 2013; Lopez and Hynes, 2003; Yue et al., 2016). Economic incentive packages launched by the central government have contributed to urban sprawl in the developing world. For instance, China took an economic reform in the 1970s by opening up land markets and commercialising housing units. This economic stimulus gave rise to many sprawling mega-cities such as Beijing, Shanghai, and Chengdu. In Europe and  North America, though, microeconomic theories may majorly explain urban sprawl. For example, households begin to relocate to the suburbs when the land prices in city centres become prohibitively high. Their relocation decisions are further strengthened by housing and land development policies. Developers promote low-density communities in the city periphery. Local governments help  build large retail centres to accommodate the increased demand. Motorization policies and low fuel costs result in automobile-oriented cities. Even public transit policies aggravate outward city growth by charging long-distance commuters less than the riders for short distances (De Vos and Witlox, 2013).

As urban sprawl increasingly threatens public health, social equity, and the built environments, people start to develop different urban containment policies. There are primarily two forms of containment policies that were adopted in the US. The State law in Oregon and Washington requires that local land-use plans should clearly define an urban growth boundary. In other states such as Florida and Maryland, governments develop urban service limits, public facilities ordinances, and other policies to promote

compact urban forms (Aytur et al., , 2008). However, the effects of urban containment policies have been hotly debate. For example, not all urban growth boundaries significantly affect housing markets and the rates of urbanisation  (Dempsey and Plantinga, 2013). Thus, De Vos and Witlox (2013) suggest the integration of spatial planning policies, mobility policies, and road pricing. Spatial planning can strictly limit new developments outside urban areas. Transit-Oriented Development benefits nurturing high-density and mixed land-use neighbourhoods. Lastly, road pricing increases long-distance travel costs, thereby curtailing urban sprawl.

The modelling approach and results offered by this work could aid in the development of an integral land-use enforcement system. Building a comprehensive land use policy for the urban growth landscape is a recommended option in coastal communities, which is currently lacking in existing principal planning practices for future SLR in the US (Fu et al., 2017). An integrative policy framework can coordinate the increased demand for urbanisation and the goal for hazard mitigation. In other words, planners ought to incorporate the  zoning of existing coastal areas  into the adaptation strategies to SLRural land us regulations to attract new development inland. While prohibiting developments within flooding zones partly  constrains urban growth and may be unrealistic, only relying on  zoning could lead to massive areas subject to flooding. A compromise of these two alternatives, accordingly, could be developed to both ensure adequate urbanisation and steer new developments away from low-lying areas. Finally, planners should also consider seawall, planned retreat, and other adaptation strategies as key components of the proposed framework  to protect existing urban areas from being inundated. Furthermore, in the long term, policy makers should formulate adaptation plans that address other SLR aspects such as groundwater pollution and saltwater intrusion beneath protective structures.

**5 Conclusions**

Environmental and resource pressures are  intensified given ongoing coastal urbanisation. Besides, the urbanisation process amplifies the exposure of coastal communities to flooding hazards. Unfortunately, we are uncertain about the degree to which SLR may contribute to the increased intensity of storminess. The possibility of more exacerbated consequences, however, cannot be neglected from a precautionary perspective. Therefore, it is crucial to building an effective coastal management plan to balance land use, competing interests, and hazard mitigation.

This work contributes to the literature by integrating urban growth dynamics, land use policies, and SLR-induced flooding. We successfully calibrated the SLEUTH model for Bay County, Florida (US), based on historical data from the year 1974 to 2013. By applying the best-fit coefficients, we developed three urban growth scenarios and assessed the exposure of future urban extent to SLR-induced flooding under different land use policies. These scenarios reflected various growth strategies that are widely applied in urban planning. Our results indicate that the parameters associated with compact development largely drive urban growth in Bay County and similar coastal communities. The results show that substantial urban growth would be prone to coastal flooding if no land use policy is implemented.

SLEUTH is particularly useful in modelling complex spatial dynamics. Moreover, the computational capacity of SLEUTH is greatly enhanced due to the rapid advancement of computer technologies. Being able to outputting GIF maps and statistics for each predictive year, SLEUTH can be easily linked with a raster-based GIS environment (Rafiee et al., 2009). Therefore, modellers can readily import the results of different scenarios  into a GIS platform for presentation purposes. Such a coupled model serves as a decision support tool and helps  land use planners, and hazard

mitigation teams evaluate the outcomes of different policies. Additionally, the visualisation results can be used to raise general awareness about the vulnerability of coastal communities to SLR.

Admittedly,  models are just   simplifications of reality, and urban growth is an intricate process that involves population increase, economic activities, and many other factors. Since the level of SLR impact on flooding is quite unclear, and growth predictions could be probably biased, the model cannot generate exact results regarding urban growth and flooding extent. However, we believe that the "what-if" estimations are useful in helping decision makers understand how policies mould distinct developments . Therefore, probabilistic models and scenario-based planning should be advocated to evaluate planning alternatives and their consequences (Xiang and Clarke, 2003) as well as to offer reliable estimates of flooding damages.

*Acknowledgements*

The authors highly appreciate Dr Keith C. Clarke for his help throughout themodelling  process. The authors greatly appreciate Dr Chih-Hung Hsu for developing flooding maps. The authors also thank the U.S. Census Bureau, the Florida Geographic Data Library, the Bay County Online, and the Bureau of Economic and Business Research for offering access to their data. This paper was undertaken with the support from the Florida Sea Grant, Grant No. R/GOM-RP-2, "A Parameterized Climate Change Projection Model for Hurricane Flooding, Wave Action, Economic Damages, and Population Dynamics". This paper was also funded by the Florida Sea Grant Project entitled "A Spatial-Temporal Econometric Model to Estimate Costs and Benefits of Sea-Level Rise Adaptation Strategies". This work received financial support from the University of Florida Graduate School Dissertation Award, and the publication of this article was funded in part by the University of Florida Open Access Publishing Fund. Last not but the least, the authors' gratitude is given to the editor, Dr Nadia Pinardi, and to two anonymous reviewers who offered constructive comments to help improve this paper.

**References**

Akın, A., Clarke, K. C., & Berberoglu, S. (2014). The impact of historical exclusion on the calibration of the SLEUTH urban growth model. *International Journal Of Applied Earth Observation And Geoinformation, 27, Part B*(0), 156-168. doi:http://dx.doi.org/10.1016/j.jag.2013.10.002

Apel, H., Trepat, O. M., Hung, N. N., Chinh, D. T., Merz, B., & Dung, N. V. (2016). Combined fluvial and pluvial urban flood hazard analysis: concept development and application to Can Tho city, Mekong Delta, Vietnam. *Natural Hazards and Earth System Sciences, 16*(4), 941-961. doi:10.5194/nhess-16-941-2016

Aytur, S. A., Rodriguez, D. A., Evenson, K. R., & Catellier, D. J. (2008). Urban Containment Policies and Physical Activity: A Time–Series Analysis of Metropolitan Areas, 1990–2002. *American Journal of Preventive Medicine, 34*(4), 320-332. doi:http://dx.doi.org/10.1016/j.amepre.2008.01.018

Bay County Online. (2016). Future land use and zoning. Retrieved from http://www.baycountyfl.gov/gis.php

Berberoğlu, S., Akın, A., & Clarke, K. C. (2016). Cellular automata modeling approaches to forecast urban growth for adana, Turkey: A comparative approach. *Landscape and Urban Planning, 153*, 11-27. doi:http://dx.doi.org/10.1016/j.landurbplan.2016.04.017

Box, G. E., & Draper, N. R. (1987). *Empirical model-building and response surfaces* (Vol. 424): Wiley New York.

Bureau of Economic and Business Research. (2016). Total population in Bay County. Retrieved from https://www.bebr.ufl.edu/population

Chakraborty, A., Wilson, B., & Kashem, S. b. (2015). The pitfalls of regional delineations in land use modeling: Implications for Mumbai region and its planners. *Cities, 45*, 91-103. doi:http://dx.doi.org/10.1016/j.cities.2015.03.008

Clarke, K. C., Hoppen, S., & Gaydos, L. (1997). A Self-Modifying Cellular Automaton Model of Historical Urbanization in the San Francisco Bay Area. *Environment and Planning B: Planning and Design, 24*(2), 247-261. doi:10.1068/b240247

De Vos, J., & Witlox, F. (2013). Transportation policy as spatial planning tool; reducing urban sprawl by increasing travel costs and clustering infrastructure and public transportation. *Journal Of Transport Geography, 33*, 117-125. doi:http://dx.doi.org/10.1016/j.jtrangeo.2013.09.014

Dempsey, J. A., & Plantinga, A. J. (2013). How well do urban growth boundaries contain development? Results for Oregon using a difference-in-difference estimator. *Regional Science and Urban Economics, 43*(6), 996-1007. doi:http://dx.doi.org/10.1016/j.regsciurbeco.2013.10.002

Dezhkam, S., Amiri, B. J., Darvishsefat, A. A., & Sakieh, Y. (2013). Simulating the urban growth dimensions and scenario prediction through sleuth model: a case study of Rasht County, Guilan, Iran. *GeoJournal, 79*(5), 591-604. doi:10.1007/s10708-013-9515-9

Dietzel, C., & Clarke, K. C. (2007). Toward optimal calibration of the SLEUTH land use change model. *Transactions in GIS, 11*(1), 29-45. doi:10.1111/j.1467-9671.2007.01031.x

Ebert, K., Ekstedt, K., & Jarsjo, J. (2016). GIS analysis of effects of future Baltic sea level rise on the island of Gotland, Sweden. *Natural Hazards and Earth System Sciences, 16*(7), 1571-1582. doi:10.5194/nhess-16-1571-2016

Felsenstein, D., & Lichter, M. (2014). Introduction to the special issue on simulating the dynamics of land use change in coastal areas. *Ocean & Coastal Management, 101, Part B*, 61-62. doi:http://dx.doi.org/10.1016/j.ocecoaman.2014.09.014

Florida Geographic Data Library. (2016). Retrieved from http://www.fgdl.org/metadataexplorer/explorer.jsp

Fu, X., Gomaa, M., Deng, Y., & Peng, Z.-R. (2017). Adaptation planning for sea level rise: a study of US coastal cites. *Journal of Environmental Planning and Management* *http://dx.doi.org/10.1080/09640568.2016.1151771*.

Garcia, E. S., & Loáiciga, H. A. (2014). Sea-level rise and flooding in coastal riverine flood plains. *Hydrological Sciences Journal, 59*(1), 204-220. doi:10.1080/02626667.2013.798660

Goldstein, N. C., Candau, J. T., & Clarke, K. C. (2004). Approaches to simulating the "March of Bricks and Mortar". *Computers, Environment and Urban Systems, 28*(1–2), 125-147. doi:http://dx.doi.org/10.1016/S0198-9715(02)00046-7

Hansen, H. S. (2010). Modelling the future coastal zone urban development as implied by the IPCC SRES and assessing the impact from sea level rise. *Landscape and Urban Planning, 98*(3–4), 141-149. doi:http://dx.doi.org/10.1016/j.landurbplan.2010.08.018

Herold, M., Goldstein, N. C., & Clarke, K. C. (2003). The spatiotemporal form of urban growth: measurement, analysis and modeling. *Remote Sensing of Environment, 86*(3), 286-302. doi:http://dx.doi.org/10.1016/S0034-4257(03)00075-0

Hsu, C.-H. (2014). *Hurricane Surge Flooding Damage Assessment and Web-Based Game Development to Support K12 Education for Understanding Climate Change Impact on Hurricane Surge Flooding Damage.* (Doctoral dissertation), Texas A & M University. Retrieved from http://hdl.handle.net/1969.1/153484

Hurricanecity. (2015). The history with tropical systems in Panama City, Florida. ——Retrieved from http://www.hurricanecity.com/city/panamacity.htm

IPCC. (2007). Climate change 2007: the physical science basis. *Summary for Policymakers*.

IPCC. (2013). *Summary for Policymakers. In: Climate Change 2013: The Physical Science Basis. Contribution of Working Group I to the Fifth Assessment Report of the Intergovernmental Panel on Climate Change [Stocker, T.F., D. Qin, G.-K. Plattner, M. Tignor, S.K. Allen, J. Boschung, A. Nauels, Y. Xia, V. Bex and P.M. Midgley (eds.)].* Cambridge, United Kingdom and New York, NY, USA: Cambridge University Press.

Irish, J. L., Resio, D. T., & Cialone, M. A. (2009). A surge response function approach to coastal hazard assessment. Part 2: Quantification of spatial attributes of response functions. *Natural Hazards, 51*(1), 183-205.

Jantz, C. A., Goetz, S. J., Donato, D., & Claggett, P. (2010). Designing and implementing a regional urban modeling system using the SLEUTH cellular urban model. *Computers, Environment and Urban Systems, 34*(1), 1-16. doi:http://dx.doi.org/10.1016/j.compenvurbsys.2009.08.003

Knutson, T. R., & Tuleya, R. E. (2004). Impact of CO2-induced warming on simulated hurricane intensity and precipitation: Sensitivity to the choice of climate model and convective parameterization. *Journal of Climate, 17*(18), 3477-3495.

Leao, S., Bishop, I., & Evans, D. (2004). Simulating Urban Growth in a Developing Nation's Region Using a Cellular Automata-Based Model. *Journal of Urban Planning and Development, 130*(3), 145-158. doi:10.1061/(ASCE)0733-9488(2004)130:3(145)

Leão, S., Bishop, I., & Evans, D. (2004). Spatial–temporal model for demand and allocation of waste landfills in growing urban regions. *Computers, Environment and Urban Systems, 28*(4), 353-385. doi:http://dx.doi.org/10.1016/S0198-9715(03)00043-7

Lopez, R., & Hynes, H. P. (2003). Sprawl In The 1990s: Measurement, Distribution, and Trends. *Urban Affairs Review, 38*(3), 325-355. doi:10.1177/1078087402238805

Ludy, J., & Kondolf, G. M. (2012). Flood risk perception in lands "protected" by 100-year levees. *Natural Hazards, 61*(2), 829-842. doi:10.1007/s11069-011-0072-6

Mahiny, A., & Gholamalifard, M. (2007). Dynamic spatial modeling of urban growth through cellular automata in a GIS environment.

Nicholls, R. J., & Cazenave, A. (2010). Sea-level rise and its impact on coastal zones. *Science, 328*(5985), 1517-1520.

Onsted, J., & Clarke, K. C. (2012). The inclusion of differentially assessed lands in urban growth model calibration: a comparison of two approaches using SLEUTH. *International Journal of Geographical Information Science, 26*(5), 881-898. doi:10.1080/13658816.2011.617305

Onsted, J. A., & Chowdhury, R. R. (2014). Does zoning matter? A comparative analysis of landscape change in Redland, Florida using cellular automata. *Landscape and Urban Planning, 121*(0), 1-18. doi:http://dx.doi.org/10.1016/j.landurbplan.2013.09.007

Onsted, J. A., & Clarke, K. C. (2011). Forecasting Enrollment in Differential Assessment Programs Using Cellular Automata. *Environment and Planning B: Planning and Design, 38*(5), 829-849. doi:10.1068/b37010

Project Gigalopolis. (2016). SLEUTH Applications Retrieved from http://www.ncgia.ucsb.edu/projects/gig/Repository/SLEUTHapplications.html

Rafiee, R., Mahiny, A. S., Khorasani, N., Darvishsefat, A. A., & Danekar, A. (2009). Simulating urban growth in Mashad City, Iran through the SLEUTH model (UGM). *Cities, 26*(1), 19-26. doi:http://dx.doi.org/10.1016/j.cities.2008.11.005

Rienow, A., & Goetzke, R. (2015). Supporting SLEUTH – Enhancing a cellular automaton with support vector machines for urban growth modeling. *Computers, Environment and Urban Systems, 49*(0), 66-81. doi:http://dx.doi.org/10.1016/j.compenvurbsys.2014.05.001

Runyan, R. C. (2006). Small business in the face of crisis: Identifying barriers to recovery from a natural disaster. *Journal of Contingencies and Crisis Management, 14*(1), 12-26.

Sakieh, Y., Salmanmahiny, A., Jafarnezhad, J., Mehri, A., Kamyab, H., & Galdavi, S. (2015). Evaluating the strategy of decentralized urban land-use planning in a developing region. *Land Use Policy, 48*, 534-551. doi:http://dx.doi.org/10.1016/j.landusepol.2015.07.004

Santé, I., García, A. M., Miranda, D., & Crecente, R. (2010). Cellular automata models for the simulation of real-world urban processes: A review and analysis. *Landscape and Urban Planning, 96*(2), 108-122. doi:http://dx.doi.org/10.1016/j.landurbplan.2010.03.001

Sekovski, I., Armaroli, C., Calabrese, L., Mancini, F., Stecchi, F., & Perini, L. (2015). Coupling scenarios of urban growth and flood hazards along the Emilia-Romagna coast (Italy). *Natural Hazards and Earth System Sciences, 15*(10), 2331-2346. doi:10.5194/nhess-15-2331-2015

Shows, E. W. (1978). Florida's coastal setback line—an effort to regulate beachfront development. *Coastal Management, 4*(1-2), 151-164.

Silva, E. A., & Clarke, K. C. (2002). Calibration of the SLEUTH urban growth model for Lisbon and Porto, Portugal. *Computers, Environment and Urban Systems, 26*(6), 525-552. doi:http://dx.doi.org/10.1016/S0198-9715(01)00014-X

Song, J., Peng, Z.-R., Zhao, L., & Hsu, C.-H. (2016). Developing a theoretical framework for integrated vulnerability of businesses to sea level rise. *Natural Hazards*, 84(2), 1219-1239. doi:10.1007/s11069-016-2483-x 1-21. doi:10.1007/s11069-016-2483-x

U.S. Census Bureau. (2016). TIGER/Line shapefiles and TIGER/Line Files. Retrieved from http://www.census.gov/geo/maps-data/data/tiger-line.html

Udoh IE (2012). Robust hurricane surge response functions. Texas A&M University, College Station

Udoh, I. E. (2012). *Robust hurricane surge response functions*: TEXAS A&M UNIVERSITY.

Vousdoukas, M. I., Voukouvalas, E., Mentaschi, L., Dottori, F., Giardino, A., Bouziotas, D., . . . Feyen, L. (2016). Developments in large-scale coastal flood hazard mapping. *Natural Hazards and Earth System Sciences, 16*(8), 1841-1853. doi:10.5194/nhess-16-1841-2016

Wagner, D. F. (1997). Cellular Automata and Geographic Information Systems. *Environment and Planning B: Planning and Design, 24*(2), 219-234. doi:10.1068/b240219

Wu, X., Hu, Y., He, H. S., Bu, R., Onsted, J., & Xi, F. (2008). Performance Evaluation of the SLEUTH Model in the Shenyang Metropolitan Area of Northeastern China. *Environmental Modeling & Assessment, 14*(2), 221-230. doi:10.1007/s10666-008-9154-6

Xiang, W.-N., & Clarke, K. C. (2003). The Use of Scenarios in Land-Use Planning. *Environment and Planning B: Planning and Design, 30*(6), 885-909. doi:10.1068/b2945

Yang, X., & Lo, C. P. (2003). Modelling urban growth and landscape changes in the Atlanta metropolitan area. *International Journal of Geographical Information Science, 17*(5), 463-488. doi:10.1080/1365881031000086965

Yue, W., Zhang, L., & Liu, Y. (2016). Measuring sprawl in large Chinese cities along the Yangtze River via combined single and multidimensional metrics. *Habitat International, 57*, 43-52. doi:http://dx.doi.org/10.1016/j.habitatint.2016.06.009